# A high precision laser scanning system for measuring shape and volume of transtibial amputee residual limbs: Design and validation

**Carson O. Squibb[1], Michael L. Madigan[2], Michael K. Philen[1]***

1 Kevin T. Crofton Department of Aerospace and Ocean Engineering, Virginia Tech (Mail Code 0203), Blacksburg, VA, United States of America, 2 Grado Department of Industrial and Systems Engineering, Virginia Tech (Mail Code 0118), Blacksburg, VA, United States of America

* mphilen@vt.edu

**Data Availability Statement:** Data underlying this paper are made accessible through the Virginia Tech Data Repository at https://doi.org/10.7294/25464223.

## Abstract

Changes in limb volume and shape among transtibial amputees affects socket fit and comfort. The ability to accurately measure residual limb volume and shape and relate it to comfort could contribute to advances in socket design and overall care. This work designed and validated a novel 3D laser scanner that measures the volume and shape of residual limbs. The system was designed to provide accurate and repeatable scans, minimize scan duration, and account for limb motion during scans. The scanner was first validated using a cylindrical body with a known shape. Mean volumetric errors of 0.17% were found under static conditions, corresponding to a radial spatial resolution of 0.1 mm. Limb scans were also performed on a transtibial amputee and yielded a standard deviation of 8.1 ml (0.7%) across five scans, and a 46 ml (4%) change in limb volume when the socket was doffed after 15 minutes of standing.

## Introduction

There are an estimated 1.6 million individuals with a lower limb amputation in the United States with an additional 185,000 lower limb amputations performed every year [1]. Among the difficulties faced by amputees, residual limb volume and shape changes can induce socket discomfort, skin irritation, and potentially skin injury. Longer term volume changes, such as following the initial amputation, can exceed 30%, and may necessitate a new socket [2–4]. Shorter diurnal volume changes, commonly resulting from sitting, standing, walking, and running, can also be significant, and often require volume management such as adding liner socks [5]. Adding liner socks, however, can be disruptive to one's lifestyle and reduce the socket-limb interface. As a result, improved strategies to accommodate for diurnal limb volume changes are of interest [3].

The ability to accurately measure the volume and shape of residual limbs is important in improving our understanding of factors contributing to volume change [6] and likely in the development of strategies to reduce its adverse effects. Methods for measuring residual limb volume vary greatly, and there have been several reviews of progress in this field [5,7–9].

**Funding:** This material is based upon work supported by the National Science Foundation under Grant No. 1906132. The funders had no role in study design, data collection and analysis, decision to publish, or preparation of the manuscript.

**Competing interests:** The authors have declared that no competing interests exist.

Current methods can be classified as either contact or noncontact. Contact methods require direct contact with the limb surface and include fluid displacement [10], anthropometric measurements [10,11], bioimpedance [6,12,13], and ultrasound systems [14,15]. Conversely, noncontact methods include laser scanners [16,17], white light scanners [17,18], photogrammetry [3], magnetic resonance imaging (MRI) [19,20], and digital image correlation (DIC) [21].

White light scanners have received more attention in recent years for measuring residual limb shape and volume changes [17,18]. Dickenson et al. compared traditional plaster casting to three commonly used scanners and found that the scanners were as reliable or less reliable as the casting process [22]. Kofman et al. also investigated the usability and reliability of three scanners (Rodin4D, Omega Tracer, and Biosculptor) by staff (two physical therapists and a resident) and found that the Rodin4D was more useable and reliable when operated by staff with limited experience with the scanners [23]. Ngan et al. investigated the reliability of the Spectra scanner in capturing the shape of a transradial residual limb and found the process to be repeatable [24]. The authors did note that future work should include comparing other scanners as well as manual casting methods. Rodrigues, Oliveira, and Gama used the Sense 3D and Microsoft Kinect v2 scanners to measure the shape of the residual limb of four unilateral participants and reported relative difference between 0.16% to 19.71% for the scanners compared to manual measurements [25]. In a more recent publication by Seminati et al., the authors investigated the reliability of the Artec Eva and Omega scanners in measuring the shape of the residual limb of 10 participants and found that the shape of the limb of the leading contributor of error in the volume measurements [26]. The authors noted that errors greater than 10% were obtained in the cross-sectional area measurements and greater than 5% for the perimeter measurements. Lewis et al. used the Artec Leo 3D scanner to compare the scanning method to the traditional water displacement test in measurement of leg volume [27]. They found that the 3D scanning had a smaller systematic bias and narrower limits of agreement than the water displacement test, however, the authors did note that the system could potentially be used to identify lymphedema but may not be able to determine leg volume changes following eccentric exercise. Nagarajan et al. used EinScan Pro 3D scanner to track the shape changes during the iterative shaping of sockets by clinicians and noted that significant alterations in the mid-patella region followed by the patellar tendon are observed [28]. The authors suggest the data can be used as training resource for junior prosthetists and help prosthetists to identify regions for rectification. One shortcoming of many of these 3D scanners noted above is that they rely on inertial measurement units (IMU) and therefore the systems are subject to drift over a period of time. Also, difficulties are experienced when scanning abrupt changes in the surface such as edges, spikes, and corners.

Three important considerations for any residual limb volume measurement system are repeatability, limb motion, and scan time. The repeatability of volume measurements can affect the reported accuracy and precision of these systems. Depending on the method, significant variably may be introduced by different users, or by the same user during different scans [10]. Among existing methods, those in which the user must manually move sensors, such as in handheld white light and laser scanners, are expected to be most susceptible to this source of error since they introduce inter-user variability. Designing a scanning device that minimizes this variability is desired [5]. Accounting for limb motion during a volume scan is a second consideration that can have a large effect on accuracy and repeatability [14,29–31]. Mitigating this effect has been attempted through by estimating and correcting for limb motion [14,15,20], decreasing scan time [21], and restricting limb motion [3,31]. However, the feasibility of implementing each of these techniques is largely dependent on the volume measurement method. Scan time is a third important consideration for limb volume measurement systems. Most volume measurement methods (other than bioimpedance) require the limb to

be outside of the socket. This can be problematic because as soon as the limb is removed from the socket, volume and shape changes occur with volume changes as high as 10% occurring within 1 minute of socket doffing [3]. Therefore, if measuring transient volume changes after doffing a socket, measurements must be completed as quickly as possible to minimize volumetric error. Scan times can vary widely depending on the method, user, and size of limb, with times being on the order of 10-60s for white light and laser scanners [17], and as short as less than one second for DIC measurements [21]. MRI and fluid displacement methods are the slowest, scan times on the order of minutes [20].

To overcome the limitations of the current scanning methods, such as slow scan times, inter-user variability, inadequate accuracy and/or repeatability, and limb motion error, the goal of this work was to develop and validate a novel high precision laser scanning system to measure limb volume and shape of transtibial amputees. Repeatability of this system is enhanced by using linear and rotary stages to automate the motion of the laser scanner over the surface of the limb and removing the potential for undesirable variations in use by human users, such as in the case of white light scanners. Additionally, limb motion during a scan is accounted for using dual camera motion tracking system to estimate and correct for the limb motion in the measured surface data. The system was first validated using a cylindrical limb model under static and dynamic testing conditions, and then using a human participant to demonstrate its ability to measure transient residual limb volume after doffing a socket.

## Research method

### Scanner design

Measurements of the limb's surface are made using a 20 mW laser light scanner (Micro-Epsilon scanCONTROL 2500–100). The scanner measures distance at 32 Hz along a laser line approximately 100 mm long with 640 data points along its length, has a rated resolution of 12 microns for surfaces 120–360 mm from the scanner. The laser sensor is mounted to an armature (Fig 1) on a custom built rotary stage, which itself is mounted onto a set of linear stages (Newport Model M-IMS600PP). The linear stages allow for the laser to be adjusted to the height of the residual limb and then be translated along the length of the limb, and whose absolute position measurements serve to define an inertial reference frame for the system. The rotary stage allows the laser head to rotate circumferentially about the limb during the scan.

At the start of each scan, the rotary and linear $z$-stage start from a fixed position at the proximal end of the limb. Prior to translation of the linear $z$-stage along the limb, one full 90°/$s$ revolution of the rotary stage is made to scan the entire circumference of the limb. While the rotary stage continues, the linear $z$-stage then translates toward the distal end of the limb at a speed that provides a 25% overlap of the laser line between consecutive rotations of the scanner, which corresponds to an effective translational velocity of 18 mm/s. To scan the distal end of the limb, the laser scanner is mounted to a miniature linear stage (TOAUTO T0601-50), referred to here as a "ministage" to avoid confusion with the two previously described linear stages. This ministage and laser scanner are mounted onto an armature which allows for the laser to be translated and rotated toward the distal end of the limb during the final portion of the limb scan (Fig 2). Once the scanner reaches the distal end of the limb, the linear $z$-stage stops translating, and the miniature linear stage translates to rotate the laser scanner. The spatial path of the laser scanner head during a complete scan is illustrated in Fig 3.

### Motion capture setup

A two-camera motion capture system was developed to measure sagittal plane limb motion during the scan to minimize motion artifact (Fig 1). Between three and five 6-mm-diameter

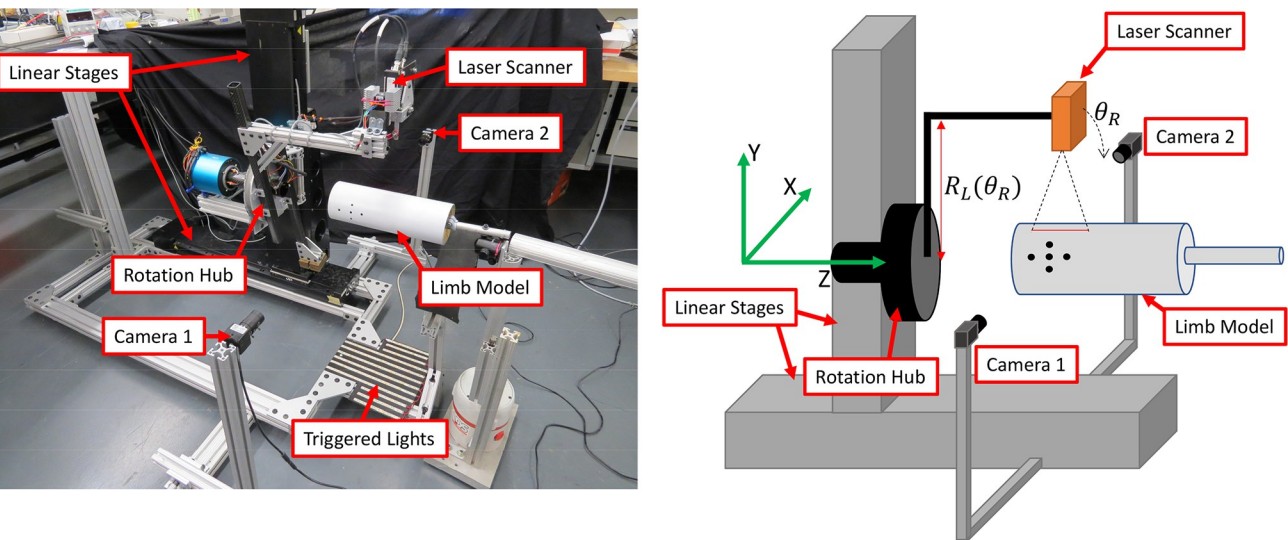

**Fig 1. Laser scanner design and depiction with 4" cylinder model in scanning area for reference.** Laser scanner, linear and rotary stages, and motion tracking hardware identified. Note that the inertial reference frame used for reconstruction of the limb is shown in the diagram on the right in green. The linear stage in the y-direction is adjusted to the height of the residual limb and the linear stage in the z-direction provides translation along the residual limb as the rotation stage (i.e. Rotation Hub) rotates the laser line sensor around the residual limb. The two-camera motion capture system measure sagittal plane limb motion during the scan to minimize motion artifacts.

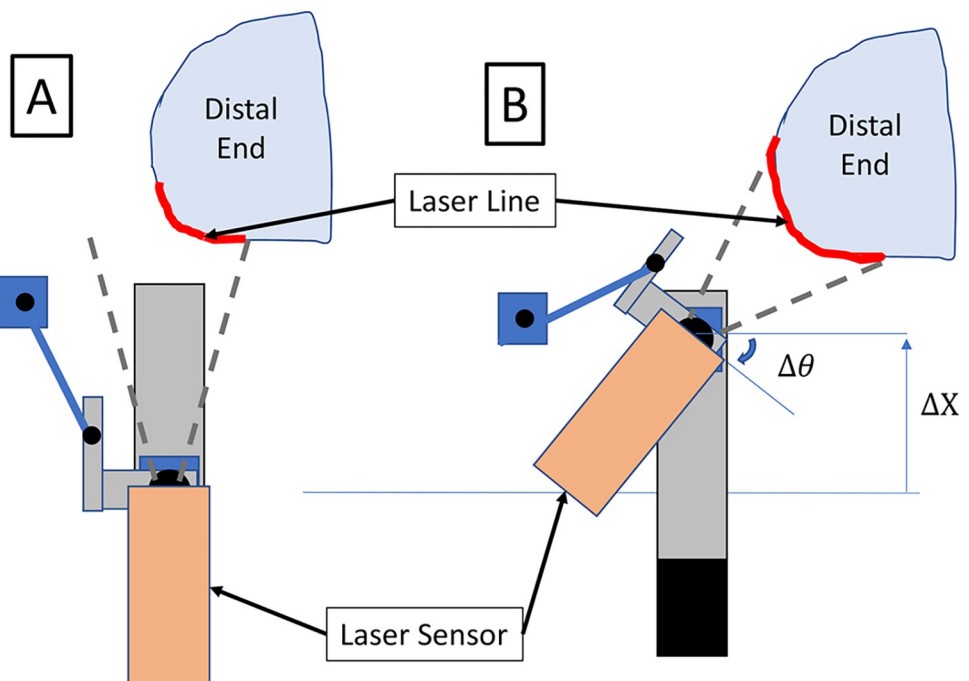

**Fig 2. Miniature linear stage, "ministage", mechanism used to translate and rotate the laser scanner toward distal end of the limb.** A) Diagram of scanner in initial orientation when laser was scanning along length of limb. B) Diagram of scanner in rotated orientation when laser was over distal end of the limb. Red lines indicate the surface of the limb scanned by the laser.

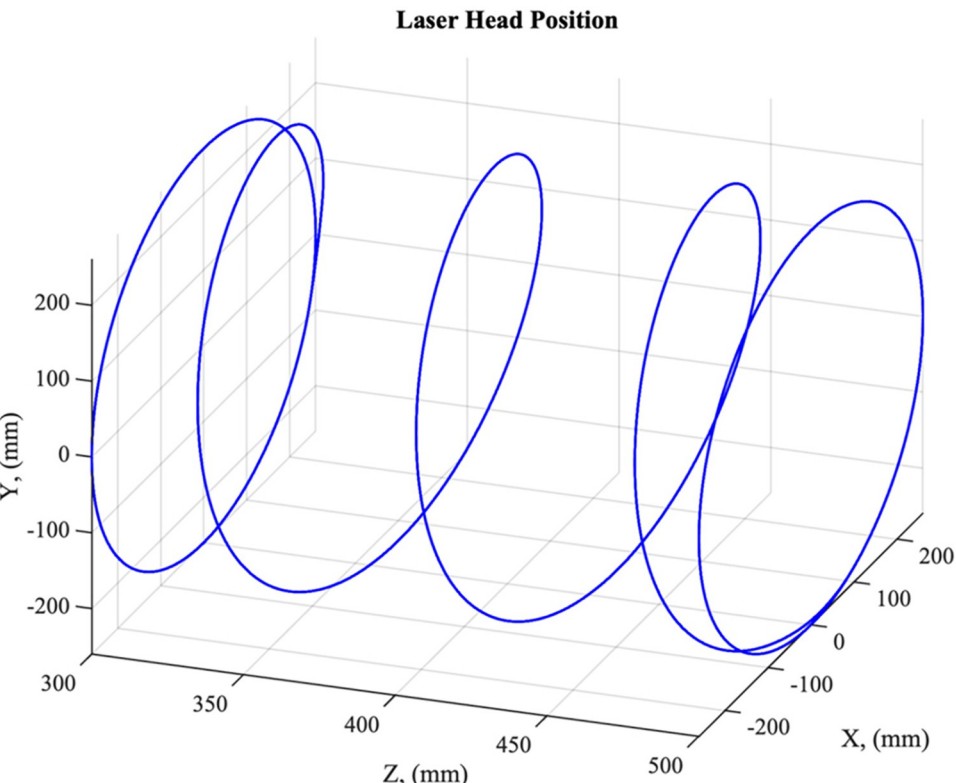

**Fig 3. Path of laser scanner during scan as it translates in the -z direction (from right to left in the diagram).** Note that at the maximum and minimum z-locations the laser completes at least one full revolution about the limb.

dot stickers are placed on the medial and lateral aspects of the distal end of the residual limb. These stickers are each tracked by 720x480 pixel cameras recording video data at 13 Hz. Two cameras were necessary due to the rotating laser scanner briefly obstructing the view of each camera of the dots for motion capture, and allows motion data to be estimated from at least one camera source throughout a scan (Fig 4). Post processing of the camera video frames (Fig 5) is then performed to add time stamps to each video frame for subsequent temporal alignment of the motion capture data and laser scanner data.

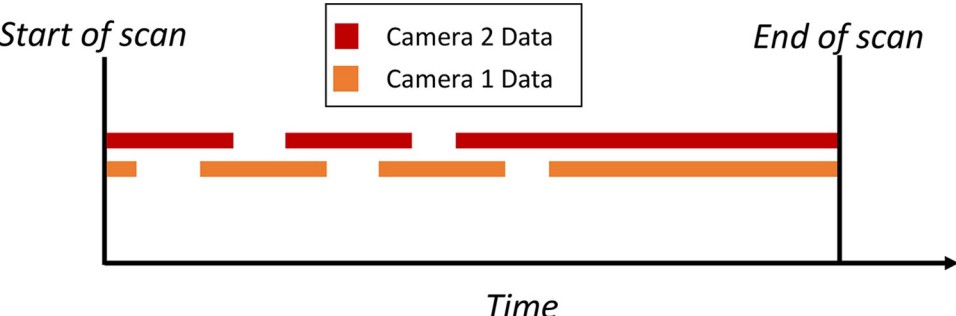

**Fig 4. Depiction of captured motion data over scan duration.** Note that, at any given time, motion data is captured for at least one camera.

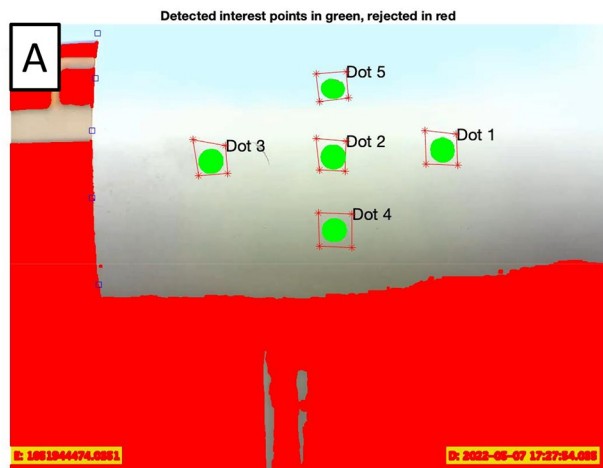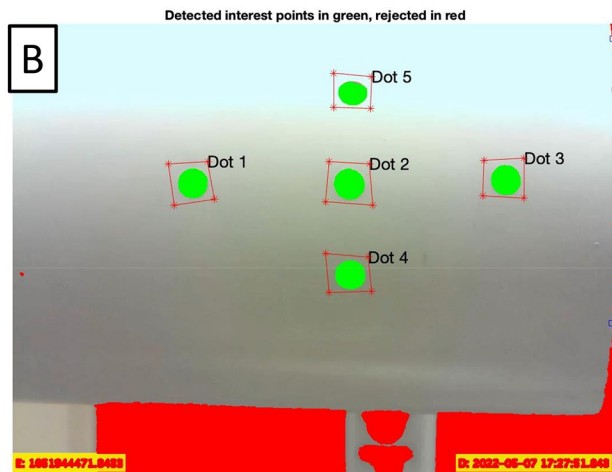

**Fig 5. First frame view of dot tracking with green regions indicating pixels that are tracked over each dot.** The labelled bounding boxes identify each dot location on the cylinder. (A) Camera 1 view. (B) Camera 2 view. Bounding boxes are defined in each image to uniquely identify the dots from one another, where the green dots indicate those to be tracked and the red areas are not tracked.

## Scanner calibration

Reconstruction of the limb surface requires the position and orientation of the laser scanner head be known in an inertially fixed reference frame throughout the scan (Fig 6). A calibration procedure was developed to determine the position and orientation of the laser scanner in this frame, while accounting for any misalignment between this frame and the laser scanner axes of motion. Additionally, compliance in the frame results in the laser calibration parameters changing as a function of it rotational position on the laser hub. For this reason, two sets of measurements are made to determine the calibration parameters of the scanning system.

First, a flat calibration object is mounted to the center of the rotation hub and fixed such that the laser line is centered over the flat surface of the object (Fig 7A), and the object's angle relative to the horizontal in the sagittal plane is measured at rotary stage angles of $0^o$, $90^o$, $180^o$, and $270^o$ using a digital level. With a 3-axis linear accelerometer (Analog Devices ADXL355) mounted to the laser sensor, 10 seconds of data is recorded at each of 21 angular positions of the laser scanner, $\theta_R$, between 0 and 360˚. Additionally, a single laser line scan is taken over the calibration object, and a picture of the calibration body from the view of the laser scanner, showing the laser line on the flat surface, is captured. This set of measurements, along with the dimensions of the calibration body, are sufficient to estimate the parameters $\phi_M$, $\psi_M$, $and\ \Delta R_M$ (depicted in Fig 7A).

Next, a flat plate is mounted to the center of the rotation hub and fixed such that the laser line is centered over the flat surface of the object (Fig 7B). With the accelerometer disconnected, a laser scan is then completed. Along with the measured orientation of the flat plate relative to the laser's axis of rotation, both the angle of the laser, $\theta_L(\theta_R)$, and the distance, or radius, of the laser this axis of rotation, $R_L(\theta_R)$, may be computed. The difference of these measurements before and after translation of the ministage during the scan also provides an estimate for the changes in this radius and angle, $\Delta R_L(\theta_R)$ and $\Delta\theta_L(\theta_R)$.

## Estimating limb motion

The two-camera motion capture data and a rigid body assumption was used to determine the rotation and translation of the limb within sagittal plane during the scan. The dot positions

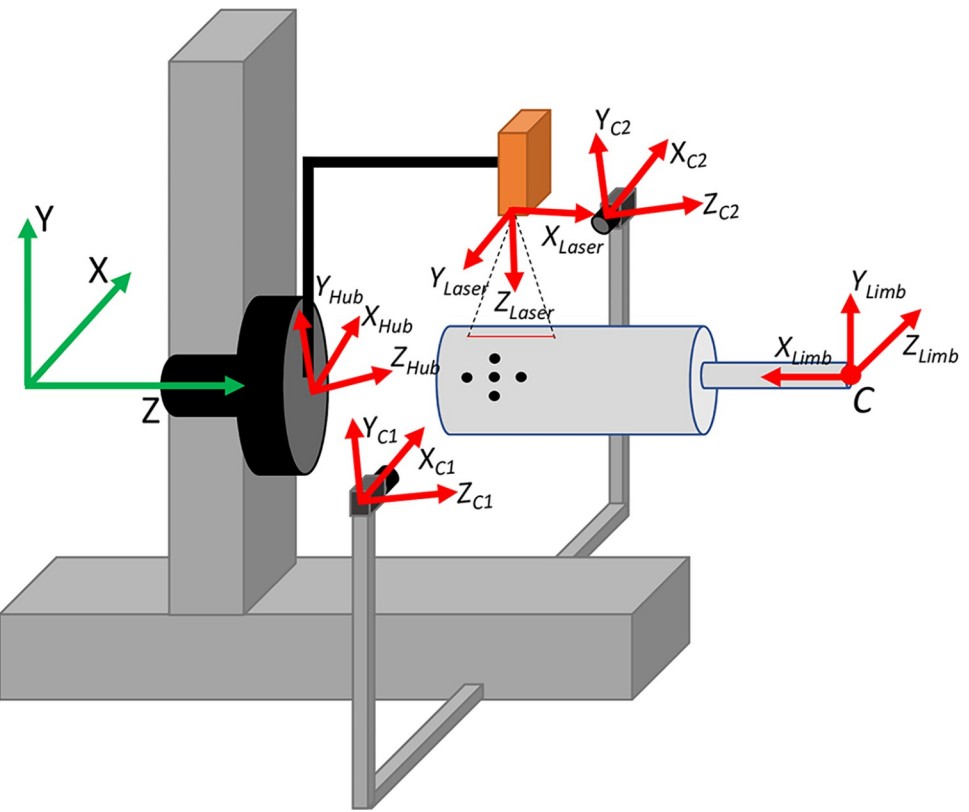

**Fig 6. Coordinate frames of the linear stages (green), rotation hub and laser, motion tracking cameras, and the limb.** Note that the center of rotation of the limb is at the knee and is indicated here by C.

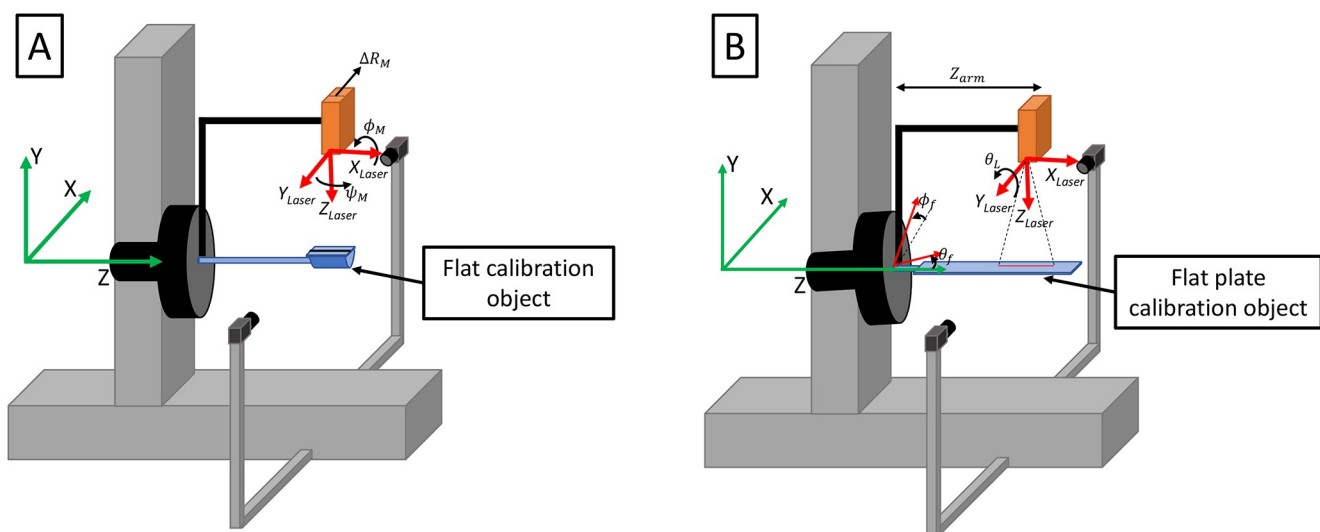

**Fig 7. Depictions of the two calibration objects in the laser scanner with relevant misalignment and calibration parameters for each case.** (A) Flat calibration object, used to estimate the two misalignment angles of the laser, $\phi_M$ and $\psi_M$, and one misalignment translation, $\Delta R_M$. (B) Flat plate calibration object, used to further estimate the laser angle, $\theta_L$, radial distance, $R_L$, and the two frame misalignment angles, $\phi_f$ and $\theta_f$.

from the motion capture cameras across video frames using the Kanade-Lucas-Tomasi tracking tool in MatLab (Fig 5). The position of the dots, in pixels, are tracked over the scan duration, and can be scaled by the known 6 mm size of the dots. The resulting dot positions can then be readily used to estimate the $x$-$y$ displacements of each dot in each camera's frame of reference, which can then be rotated to align with the limb reference frame (see Fig 6 for coordinate frame definitions). The times of frame interference for each camera is determined by using a moving standard deviation of the pixel light intensity in each dot bounding box where then a threshold is set to identify when not to track the dots. The limb's sagittal plane rotation angle is found by taking two dot pairs from the tracked dot positions and computing the angle of them to each camera's $x$-axis, relative to the first video frame.

The limb's sagittal plane center of rotation, which should be the knee joint, is found using a method developed by Spiegelman and Woo [32], where the arbitrary translation of dots on a rigid body are used to estimate the center of rotation of the body while accounting for rigid body translation as well. The method is found to work acceptably well for rotation angles greater than $10^o$, and allows for freedom of the dot placement. As described below, video recordings during suitably large rotations are obtained prior to laser scans by asking subjects to flex then extend their knee through its range of motion. With these data, the center of rotation of the limb can be determined for each camera (Fig 5). Lastly, the rigid body translations of the limb can be estimated by subtracting the total dot displacements from those due to limb rotation and averaging the results for each dot set.

## Alignment of data sets: Spatial and temporal

Three sets of data must be aligned spatially and temporally including data from the rotary and two linear stages, the laser scanner and ministage, and two video cameras. The start times of data collection for the laser scanner and all stages are recorded in epoch time using the laser scanner computer's internal clock. This same computer is used to trigger a light source positioned under the limb that provides a signal in both camera's field of view as a change in brightness. This temporal alignment carries an uncertainty of approximately 70 ms when video recording at 13 Hz, and is dependent on the frame rate of the cameras.

In addition to temporal alignment of the camera and laser scanner data sets, spatial alignment between these same two data sets is necessary so that the limb rotation and translation can be corrected for in the limb surface data. This is done by identifying the dot locations in the laser scanner data (note that these data have not been corrected for rigid body motion yet) that appear as regions of missing data due to the reflectivity of the dots (Fig 8). Using these measured dot locations, the position of the limb's center of rotation can be readily identified in the inertial reference frame of the linear stages.

## Mapping of data

Having obtained the calibration parameters for the laser scanner, the scan data from a given scan/limb surface must be mapped into the inertially fixed Cartesian coordinate frame of the linear stages (Fig 6), assuming no motion of the limb. This process involves two steps: 1) correcting the calibration parameters for any misalignment between the laser rotation hub and the inertial coordinate frame, and 2) mapping each measured limb surface point from the laser, $(x_L, z_L)$, into the coordinate frame of the laser rotation hub (see Fig 6). First, at a given rotation position, $\theta_R$, the change in the misalignment angles of the laser scanning head, $\theta_L$ and $\psi_M$, are given by Eqs (1) and (2). With these angle corrections, the corresponding changes to the radius, $R_L$, tangential radial offset, $\Delta R_M$, and $z$-position of the laser head may be derived,

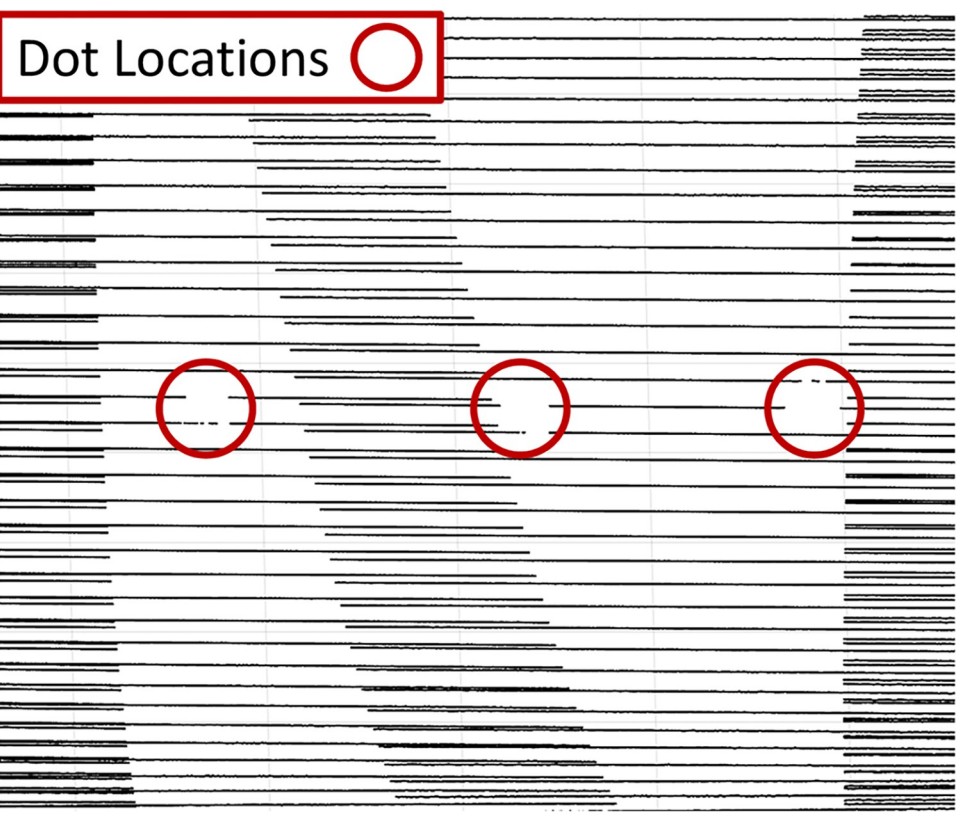

**Fig 8. Laser scanner data with regions of missing data corresponding to dot locations highlighted.**

and are given by Eqs (3–5).

$$\theta_{corr} = \theta_f \cos(\theta_R) + \phi_f \sin(\theta_R) \tag{1}$$

$$\psi_{corr} = -\theta_f \sin(\theta_R) + \phi_f \cos(\theta_R) \tag{2}$$

$$R_{corr} = Z_{arm} \sin(\theta_{corr}) - R_L(\theta_R)(1 - \cos(\theta_{corr})) \tag{3}$$

$$Z_{corr} = -Z_{arm}(1 - \cos(\theta_{corr})\cos(\psi_{corr})) - R_L(\theta_R)\sin(\theta_{corr}) \tag{4}$$

$$\Delta R_{M,corr} = Z_{arm} \sin(\psi_{corr}) \tag{5}$$

Finally, the corrected misalignment angles of the laser head at a given angular position are approximated by Eqs (6) and (7).

$$\theta_L = \theta_L + \theta_{corr} \cos(\phi_M)\cos(\psi_M) \tag{6}$$

$$\psi_M = \psi_M + \psi_{corr} \tag{7}$$

With the misalignments of the laser rotation hub and the laser scanner accounted for, the next step is mapping each measured data point from the laser scanner, $(x_L, z_L)$, into the chosen $x$-$y$-$z$ inertially fixed Cartesian coordinate frame. This mapping approach is depicted in Fig 9 for an arbitrary cross section of a limb, with the Cartesian coordinate system shown. The

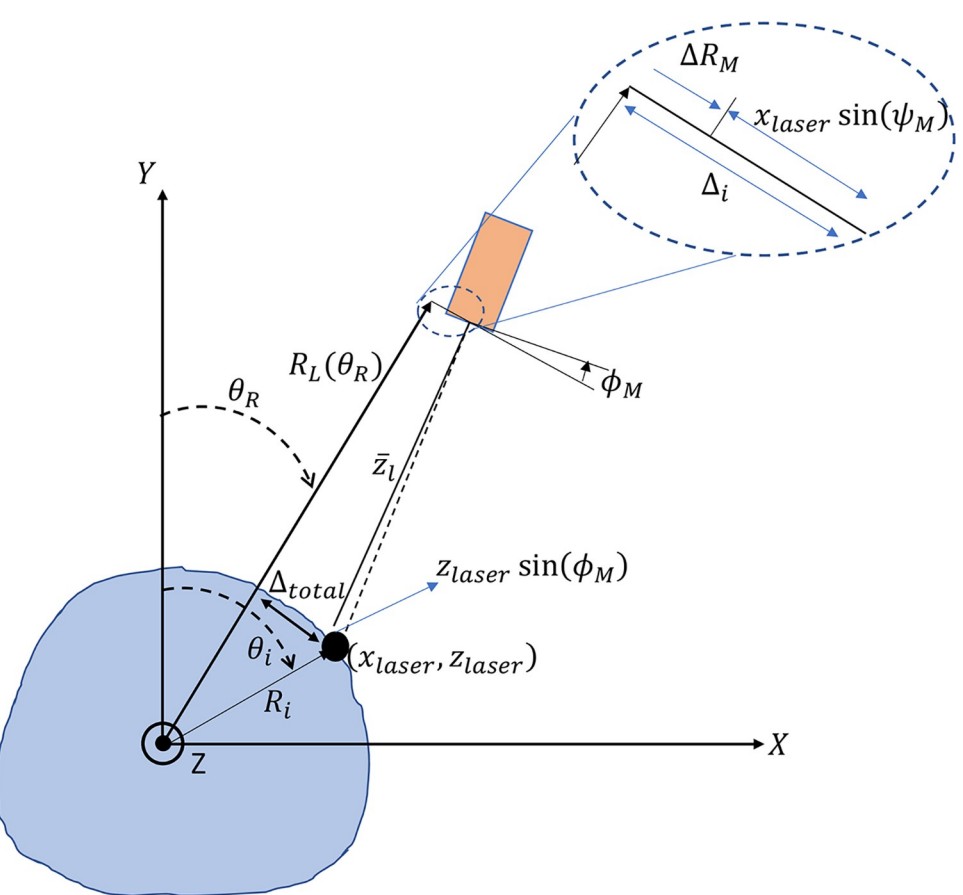

**Fig 9. Depiction of method to identify location of surface point on limb, $(R_i, \theta_i)$, in an inertially fixed cylindrical coordinate frame of the linear stages.** Here the z-axis lies on the center of rotation for the laser scanner.

method seeks to find the true angular position and radius of the current surface point on the limb, $(R_i, \theta_i)$, in an inertially fixed cylindrical coordinate system first, with a $z$-axis coinciding with that of the inertially fixed Cartesian coordinate system.

First, the current laser point may be projected onto aligned axes of Z and that of $R_L$, the radial axis of the laser head if it had zero misalignment. This aligned point location relative to the laser head, $(\bar{x}, \bar{z})$, is depicted in Fig 10 with the laser head, and the equations for this point in terms of the misalignment angles are given in Eqs (8) and (9). Next, the total offset distance of a point in the direction of rotation, $\Delta_{total}$, may be found using the misalignment rotations and translations, and is given by Eq (10).

$$\bar{x}_L = x_L \cos(\theta_L)\cos(\psi_M) + z_L \sin(\theta_L)\cos(\phi_M) \tag{8}$$

$$\bar{z}_L = z_L \cos(\theta_L)\cos(\phi_M) - x_L \sin(\theta_L)\cos(\psi_M) \tag{9}$$

$$\Delta_{tot} = \Delta R_M(\theta_R) + z_L\sin(\psi_M) + z_L\sin(\phi_M) \tag{10}$$

With this information, the equivalent radial position of the point on the limb surface, $R_i$, is found from the right triangle formed from the radial axis of the laser and this total offset distance, and is given in Eq (11). Similarly, the equivalent angular position of the limb surface

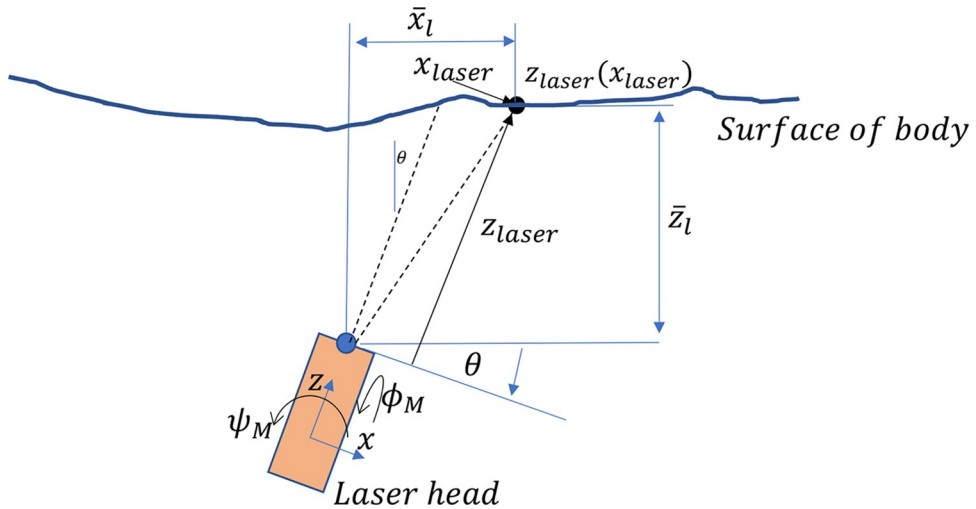

**Fig 10. Projected distance of surface point, $(\bar{x}_l, \bar{z}_l)$, from laser head.**

point, $\theta_i$, may also be found from this information. Here though there exists the possibility that the limb surface point lies either in the same or the opposing quadrant as the laser head. Equivalently, there are two cases, 1) where $R_L(\theta_R) \geq \bar{z}_L$, or 2) $R_L(\theta_R) < \bar{z}_L$, which leads to two formulae for this equivalent angle in Eq (12).

$$R_i = \sqrt{\Delta_{total}^2 + (R_L - \bar{z}_L)^2)} \tag{11}$$

$$\theta_i = \begin{cases} \theta_R + \tan^{-1}\left(\dfrac{\Delta_{total}}{R_L - \bar{z}_L}\right), & \text{if } R_L \geq \bar{z}_L \\[3mm] \theta_R + 180 - \left|\tan^{-1}\left(\dfrac{\Delta_{total}}{R_L - \bar{z}_L}\right)\right|, & \text{if } R_L < \bar{z}_L \end{cases} \tag{12}$$

Finally, with the in-plane position of the surface point on the limb determined in a cylindrical coordinate system, the point in the Cartesian coordinates, $[X_i, Y_i, Z_i]^T$, may be found using trigonometric relations as in Eq (13).

$$\begin{bmatrix} X_i \\ Y_i \\ Z_i \end{bmatrix} = \begin{bmatrix} R_i \sin(\theta_i) \\ R_i \cos(\theta_i) \\ X_{stage} + \bar{x}_L + Z_{corr} \end{bmatrix} \tag{13}$$

## Motion correction

Using the estimated rigid body motion, $(\Delta X_m, \Delta Y_m, \Delta\theta_m)$, this motion can then be corrected for in the laser scanner data. From the initially mapped scanner data, the displaced point locations, $(x, y, z)_i$, of the limb are known. With the motion data both temporally and spatially aligned with the laser scanner data, each point of the limb surface is then motion corrected. In the case of limb motion, the laser scanner data have been translated and/or rotated about the limb's center of rotation from their true position into the final measured positions. In accounting for the limb rotation, the uncorrected limb surface data must be shifted so that the data origin is at the limb's center of rotation, $[C_x, C_y, C_z]$. For a given point on the surface, $[X, Y, Z]_i$ the translated point, $[X, Y, Z]_t$, may be found using the center of rotation location using Eq

(14).

$$\begin{aligned} X_t && X_i - C_x \\ Y_t &=& Y_i - C_y \\ Z_t && Z_i - C_z \end{aligned} \tag{14}$$

Next, the Euler rotation matrix for the limb rotation through the two estimated limb rotation angles, $\theta_{axial}$ and $\theta_m$, are given by Eq's (15) and (16). Note that in each of these matrices, the negative of the estimated joint angle is required due to the difference in coordinate systems in the limb motion and the laser scanner inertial frame. The final rotation matrix that rotates an initial data point, $[X, Y, Z]_j$, is then given by Eq (17), using a 1–3 rotation sequence.

$$R_{\theta_a} = \begin{bmatrix} \cos(-\theta_{axial}) & \sin(-\theta_{axial}) & 0 \\ -\sin(-\theta_{axial}) & \cos(-\theta_{axial}) & 0 \\ 0 & 0 & 1 \end{bmatrix} \tag{15}$$

$$R_{\theta} = \begin{bmatrix} \cos(-\theta_m) & \sin(-\theta_m) & 0 \\ -\sin(-\theta_m) & \cos(-\theta_m) & 0 \\ 0 & 0 & 1 \end{bmatrix} \tag{16}$$

$$R_i = R_{\theta_a} R_{\theta} \tag{17}$$

Using these results, the relationship between the true and uncorrected surface data points, centered about the limb's center of rotation, is readily found in terms of the limb rotation matrix and the previously estimated limb translations, $\Delta Y_m$ and $\Delta X_m$, in Eq (18).

$$\begin{bmatrix} X_t \\ Y_t \\ Z_t \end{bmatrix} = R_i \begin{bmatrix} X_j \\ Y_j \\ Z_j \end{bmatrix} + \begin{bmatrix} 0 \\ \Delta Y_m \\ -\Delta X_m \end{bmatrix} \tag{18}$$

Finally, the motion corrected true surface data, $[X, Y, Z]_0$, is found by translating the data back into the inertial coordinate frame, as in Eq (19)

$$\begin{aligned} X_0 && X_j + C_x \\ Y_0 &=& Y_j + C_y \\ Z_0 && Z_j + C_z \end{aligned} \tag{19}$$

## Model cylinder scans

Validation of the laser scanner was completed using a high tolerance, 101.6 mm nominal diameter aluminum cylinder that was representative of a transtibial residual limb. The cylinder was painted matte white to mitigate surface reflectivity with the laser scanner, and two 6 mm dot sets were applied to the lateral and medial aspects of the distal end of the cylinder (Fig 1). Vernier calipers were used to measure a true diameter of 102.25±0.072 mm. Uncertainty estimations of the laser calibration method were made by performing three calibrations. Each calibration included a minimum of four flat plate calibration scans, which allowed for

measurements of uncertainty for each calibration. After each calibration, one set of five cylinder scans were completed under static conditions (i.e., stationary) and another set of five scans under dynamic conditions with the cylinder moving to quantify the effectiveness of the motion correction procedure. As reported elsewhere for other limb scanners [8,18,33], the reliability of the scanner during limb model scans was quantified using typical error of the measurement (TEM) calculated as the standard deviation of volume measurements divided by $\sqrt{2}$, ICC(3,1), and a reliability coefficient was calculated as $1.96(\sqrt{2})$TEM.

## Human participant scans

The laser scanner was also validated using a human participant volunteer. The subject was a 60 year old man (body mass = 109 kg, height = 1.72 meters) with a unilateral transtibial amputation due to diabetes that was performed 2.5 years prior. This research was approved by the Virginia Tech Institutional Review Board (VT IRB# 20–364), and the subject provided written informed consent prior to participation.

Upon arrival, the subject was asked to sit in a chair and remove his socket and liner. Following a 20-minute rest period to allow the limb volume to equilibrate, Phase 1 involved five consecutive scans without moving the limb between scans. These scans provided a baseline measure of limb resting volume, and data to quantify scanner reliability on a subject's residual limb. As reported elsewhere for other limb scanners [22,23,26], the reliability of the scanner during residual limb scans was quantified using TEM and a reliability coefficient calculated as $1.96(\sqrt{2})$TEM. Next, for Phase 2, the subject was asked to remove and reinsert his limb into the scanning area three times, each time taking one scan to give a measure of repeatability of scanning for different initial limb orientations. Next, the subject donned his socket and stood for 15-minutes. Then subject then sat again in the chair, quickly doffed his socket, and immediately inserted his limb into the scanning area. Phase 3 scans were then completed back-to-back for 15 minutes, totaling 12 scans with the first scan beginning approximately 10 seconds after doffing of the socket. This final set of scans was made to demonstrate the laser scanners' ability to measure short-term limb volume changes of a residual limb after removal from a socket. Approximately 130 mm of the residual limb length was scanned (determined by the length of the limb itself and not limited by the laser scanner). Each scan lasted 34 seconds, and consecutive scans could be started every 75 seconds after allowing for time to save the scanner data and resetting of the laser position.

## Results and discussion

The four estimated parameters from the flat plate calibration, $\theta_L(\theta_R)$, $R_L(\theta_R)$, $\Delta\theta_L(\theta_R)$, $and$ $\Delta R_L(\theta_R)$, were identified as having the greatest effect on the calibration accuracy and results. These calibration parameters can be plotted in as a function of $\theta_R$ for each calibration, and for each of the flat plate scans taken in each calibration. Doing so for each parameter, $\theta_L(\theta_R)$, $R_L(\theta_R)$, $\Delta\theta_L(\theta_R)$, $and$ $\Delta R_L(\theta_R)$, and for each calibration in Figs 11–14, respectively. Additionally, the mean value of each calibration parameter, averaged over $\theta_R$, and the mean and standard deviations, averaged over $\theta_R$, are reported in Table 1. Here the repeatability of flat plate scans within each calibration is supported by low standard deviations for each parameter, with maximum values of 0.03 mm, 0.03˚, 0.14 mm, and 0.06˚ for $\theta_L$, $R_L$, $\Delta R_L$, $and$ $\Delta\theta_L$, respectively. Further comparing the mean values between calibrations find standard deviations of the means to be 0.07 mm, 0.03˚, 0.057 mm, and 0.03˚ for the same four respective parameters. These results indicate a negligible difference in inter-calibration and intra-calibration repeatability of the laser scanner.

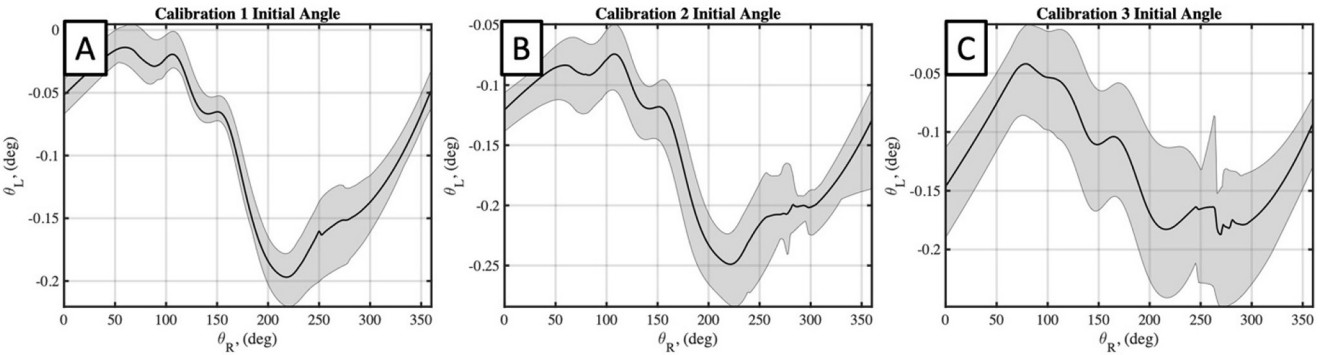

**Fig 11. Calibrated initial laser angle for three performed calibrations.** A, B, and C plots results for calibrations 1,2, and 3, respectively. Black line plots mean value of each parameter, averaged over $\theta_R$, with a grey corridor plotting minimum and maximum values over five flat plate scans.

## Static cylinder tests

Next, the static cylinder scans were compared to the caliper-based measurements. Three key metrics of comparison were made between the laser scanner data and the known shape: 1) the cylinder's diameter, 2) the cylinder surface area, and 3) the cylinder volume. While each of these three quantities are interrelated, they provide different insights to the scanning and post-processing errors, where the surface area and cylinder volume include effects from being meshed to create a closed surface to compute these measures. The limitations of using known reference shapes, and concerns regarding using scalar metrics (surface area and volume) to quantify measurement accuracy have been acknowledged previously [5], so the spatial accuracy of the measured cylinder diameters were also computed to mitigate these concerns.

The cylinder diameter as measured by the laser scanner was determined by taking cross sections of the reconstructed surface along the $z$-axis of the inertial reference frame. For each cross section, diameter measurements were taken circumferentially within the $x$-$y$ plane, allowing for the accuracy of the laser scanner to be measured spatially about the circumference of the cylinder and along the scanned length. The scanned cylinder surface area and volume were obtained by meshing the scanned point cloud data in MeshLab. The raw data, typically containing over 1 million data points, was first down sampled to 100,000 points before being meshed using a screened Poisson's surface reconstruction method (Fig 15). From the reconstructed cylinder surface, the meshed surface area and volume was computed and used for comparison to the expected values from the known geometry.

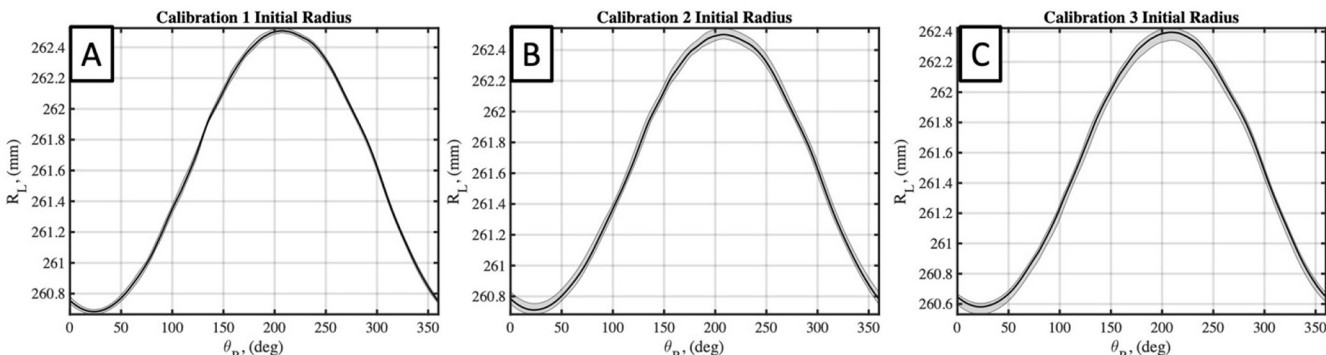

**Fig 12. Calibrated initial laser head radius for three performed calibrations.** A, B, and C plots results for calibrations 1,2, and 3, respectively. Black line plots mean value of each parameter, averaged over $\theta_R$, with a grey corridor plotting minimum and maximum values over five flat plate scans.

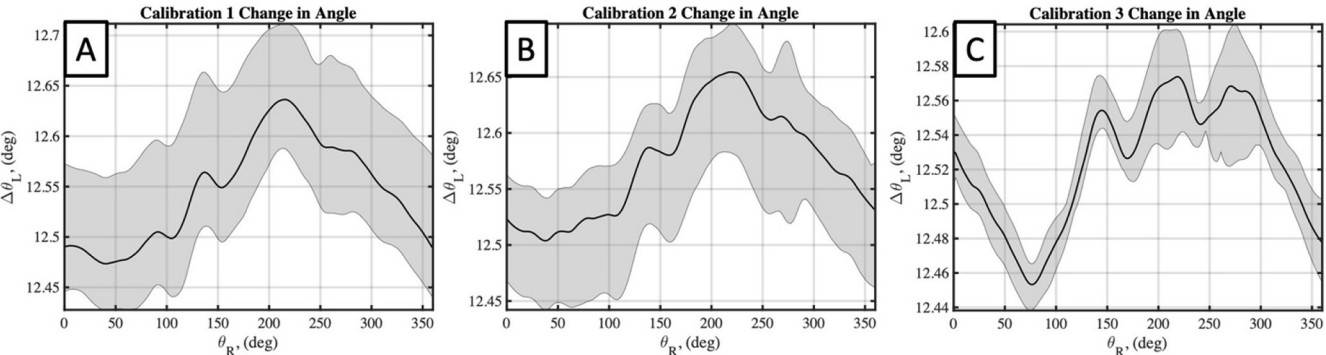

**Fig 13. Calibrated change in laser angle for three performed calibrations.** A, B, and C plots results for calibrations 1,2, and 3, respectively. Black line plots mean value of each parameter, averaged over $\theta_R$, with a grey corridor plotting minimum and maximum values over five flat plate scans.

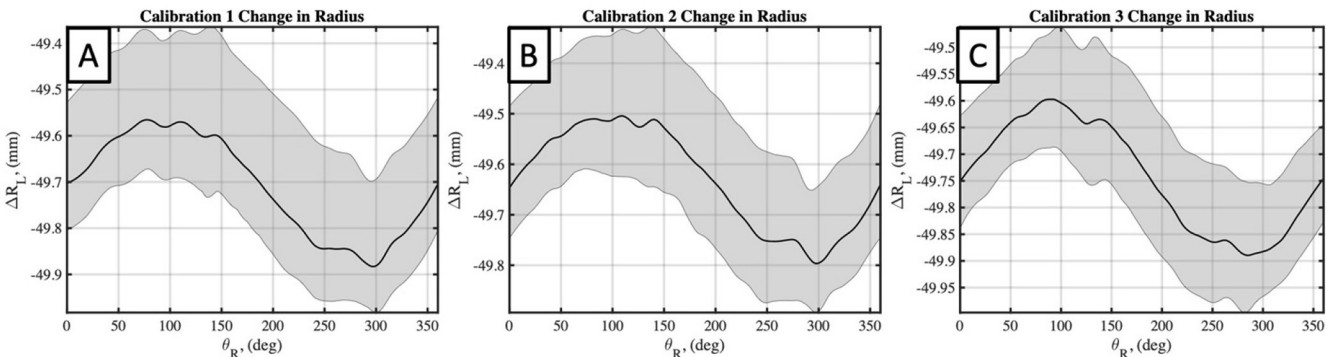

**Fig 14. Calibrated change in laser head radius for three performed calibrations.** A, B, and C plots results for calibrations 1,2, and 3, respectively. Black line plots mean value of each parameter, with great corridor plotting minimum and maximum values.

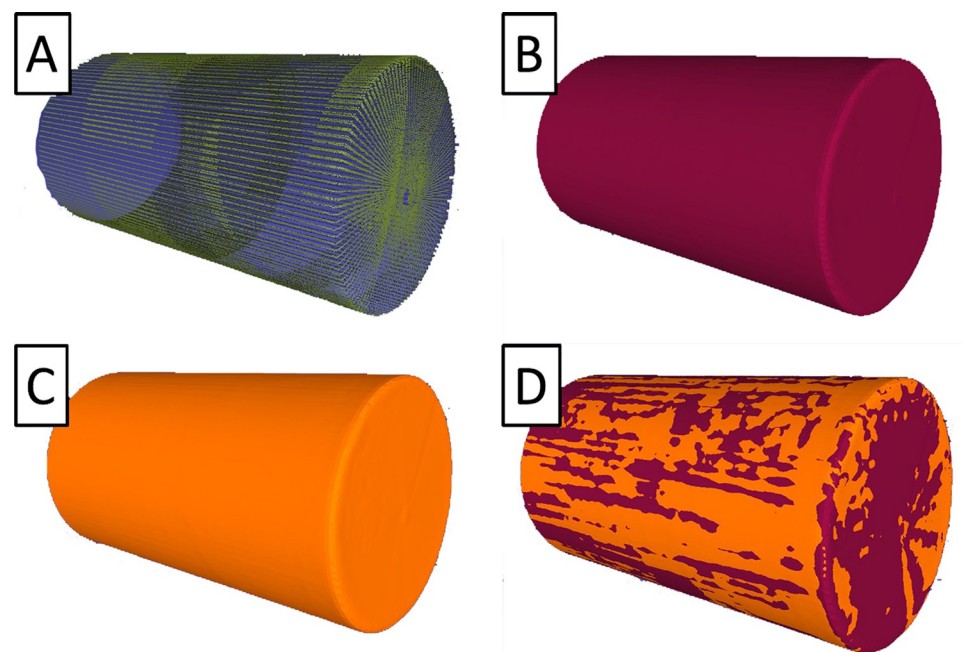

**Fig 15. Sampled and meshed static cylinder.** (A) Down sampled points cloud data from scan. (B) Corresponding meshed surface. (C) Meshed surface from subsequent scan. (D) Overlay of two scans, demonstrating repeatability of reconstructed surface.

**Table 1. Laser calibration parameters for each of three calibrations, with values averaged over $\theta_R$ and standard deviations reported.**

| Calibration | $R_L(\theta_R)$ (mm) | $\theta_L(\theta_R)$ (deg) | $\Delta R_L(\theta_R)$ (mm) | $\Delta\theta_L(\theta_R)$ (deg) |
|---|---|---|---|---|
| 1 | 261.613 ± 0.020 | -0.09±0.02 | -49.71 ± 0.14 | 12.55±0.06 |
| 2 | 261.627 ± 0.030 | -0.15±0.02 | -49.63 ± 0.11 | 12.57±0.05 |
| 3 | 261.500 ± 0.030 | -0.12±0.03 | -49.74 ± 0.09 | 12.52±0.02 |

Cylinder scans exhibited acceptably small errors when compared to the caliper-measured diameter, with variations along the length of the cylinder and between calibrations (Fig 16; Table 2). The maximum error in cylinder diameter of any scan was measured as 0.33 mm (0.33%). The mean error from each calibration were all within 0.15 mm (0.15%) of the diameter measured manually. Overlap between subsequent cylinder meshes is illustrated in Fig 15C and 15D for two static scans, with the different colored regions in Fig 15D corresponding to visible areas of each mesh.

The cylinder volume was 1,434.3 mL and surface area was 72,534 mm$^2$ as computed with the caliper hand measurements of the cylinder diameter and assuming a circular cross section. The mean (standard deviation) cylinder volume as determined from the laser scanner during static scans was 1,437.2 (0.79) mL. The volumetric and surface area error for the static scans was deemed acceptably small (Table 2). The maximum absolute mean volumetric error (expressed related to the volume determined from caliper hand measurements) was 2.9 mL (0.2%) with a standard deviation of 0.39 mL (0.08%) among scans and a 95% confidence interval of (-0.33%, -0.08%). Similarly, the maximum absolute mean error in surface area was 5.72 cm$^2$ (0.79%) with a standard deviation of 0.60 cm$^2$ (0.08%) and a 95% confidence interval of (-0.92%, -0.66%).

## Dynamic cylinder tests

To quantify the effectiveness of motion correction, the cylinder was mounted to an electrodynamic shaker (Fig 1) to apply a 2 Hz sinusoidal excitation to the cylinder during scans. 2 Hz excitation was chosen as a maximum expected response from a human participant, thus providing the most challenging scenario for the motion correction algorithm. This resulted in primarily a rigid body translation of the cylinder in the $y$-direction, but compliance in the mount also resulted in translation in the $x$-direction and rotation of the cylinder about the $z$-axis. Excitation in the y-direction was chosen to best replicate the expected motion of the residual limb during scanning. From the motion estimation results, an example of the computed rigid

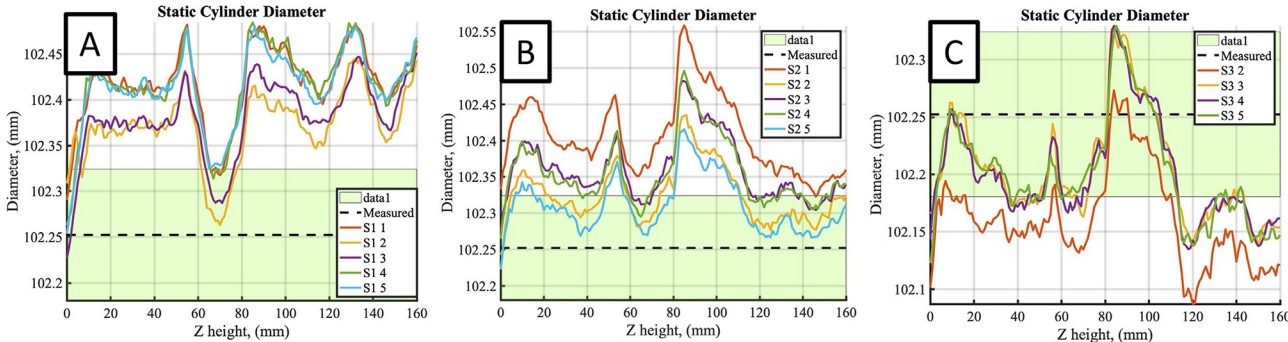

**Fig 16. Mean static cylinder diameter profiles as a function of position along length of body.** Black dashed line and green window plots the mean and standard deviation of the hand measured diameter. Plots A, B, and C plot the results for calibrations 1, 2, and 3, respectively.

**Table 2. Computed accuracy measures for static cylinder scans across three calibrations.**

| Calibration | Mean Diameter (mm) | Mean Diameter Error (mm) | Mean Diameter Error (%) | Max Diameter Error (%) | Volumetric Error (%) | Surface Area Error (%) |
|---|---|---|---|---|---|---|
| 1 | 102.400 | 0.147 | 0.144 | 0.264 | 0.18±0.03 | −0.58±0.06 |
| 2 | 102.352 | 0.099 | 0.097 | 0.325 | 0.14±0.19 | −0.65±0.22 |
| 3 | 102.192 | -0.060 | -0.059 | -0.177 | −0.20±0.08 | −0.79±0.08 |

body motion of the cylinder is plotted in Fig 17 as a function of time for each degree of freedom. The primary motion was shown to be a rigid body *y*-direction translation, with a peak to peak magnitude of approximately 0.65 mm.

Five cylinder scans were completed during the 2 Hz excitation for each calibration. The scanned cylinder diameter profiles, meshed cylinder volume, and surface area were computed both with and without motion correction. The mean diameter profiles are plotted in Fig 18 in a similar manner to those for the static test, now for the 2 Hz motion corrected scan data. The mean (standard deviation) cylinder volume as determined from the laser scanner was 1,433.7 (1.15) mL and 1439.5 (1.80) mL during uncorrected and corrected dynamic scans, respectively. Table 3 lists the meshed volume and surface area measurement errors computed from the corrected scan data. Cylinder diameter measured using the scanner after motion correction exhibited a mean error in diameter of 0.23% across all calibrations, with a maximum error of 0.51% for any individual scan. Similarly, the absolute mean volumetric and surface area errors across all calibrations was found to be 0.38% and 0.59%, respectively, with corresponding maximum absolute errors for any calibration of 0.53% and 0.87%. As is evident from the plotted diameter data, and from the reported standard deviations on the volume and surface area, a key difference between these dynamic scan tests and the previous static scans is an observed increase in variance between scanned cylinder measurements for the motion corrected results.

In comparing dynamic scans with and without motion correction, a relative change in error between these scans and the static scan data provides a measure of effectiveness of the motion correction method. This relative change in error for limb volume and surface area is computed with and without motion correction for the original and motion-corrected scanner

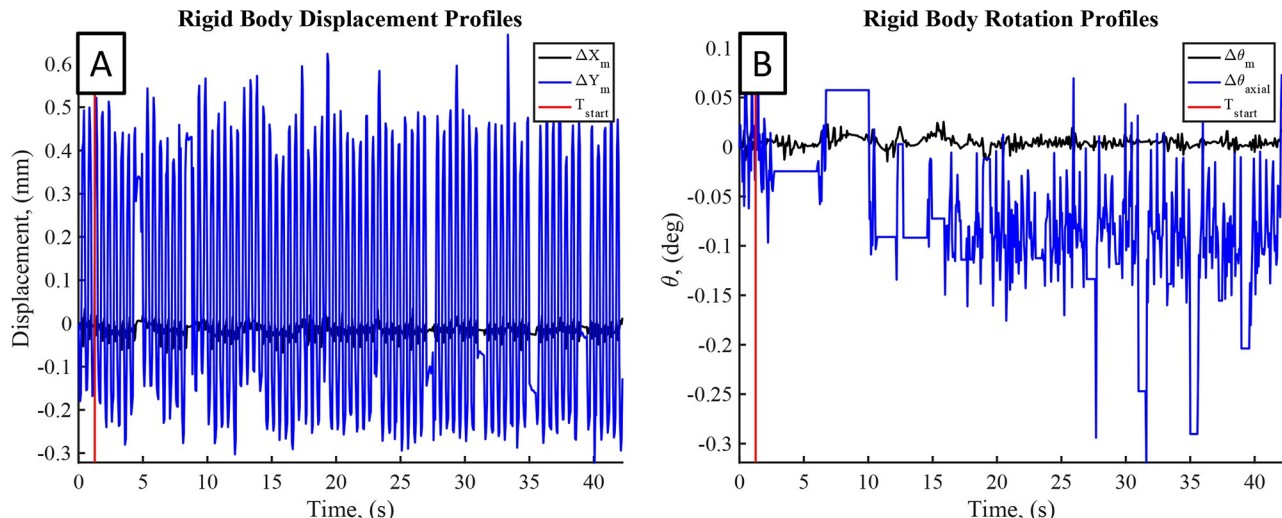

**Fig 17. Rigid body displacements and rotations for example 2 Hz excitation on 101.6 mm cylinder model.** The vertical red line indicates the start of the scanning data collection. (A) Estimated rigid body translations, and (B) Estimated rigid body rotations of model about the limb z-axis.

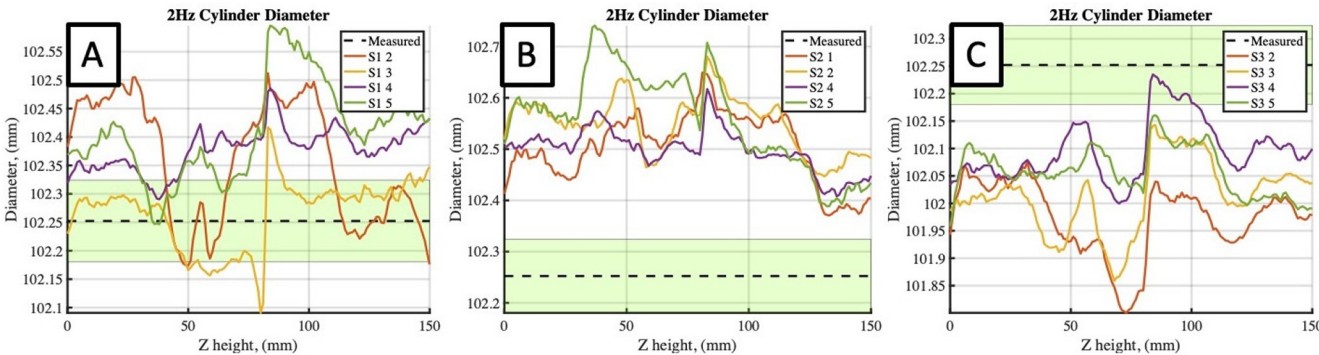

**Fig 18. Cylinder diameter profiles using scanner data during 2 Hz excitation with motion correction.** A,B,C plotted for scans from calibrations 1,2, and 3, respectively.

data (Table 4). Data from the dynamic tests resulted in a maximum change in volumetric error across all calibrations of -0.42%, with a mean change in magnitude of volume of 0.31%. Correcting for cylinder motion, however, resulted in a mean change in volumetric error of 0.20% across calibrations. The maximum improvement in volumetric error of 0.24% was observed in calibration 1. This demonstrates that, for these small magnitude but high frequency applied motions to the cylinder, the motion correction method developed for this scanning system is effective in reducing the induced error in the scan data from limb motion.

An example meshed cylinder, for static, dynamic-uncorrected, and dynamic-corrected cases, is shown in Fig 19. Here the benefit of motion correction is evident in Fig 19C for the overlaid meshed surfaces, where the uncorrected meshed surface (maroon) is found to have an induced radial, periodic error when compared to the motion corrected (orange) mesh. Although the cylinder surface area and volume may indicate a small change in the cylinder shape, the meshed shapes are different as a result of the applied motion.

The reliability of the volume measurements as quantified using TEM was 0.56 mL for static scans, 0.81 mL for dynamic scans uncorrected for motion, and 1.27 mL for dynamic scans after correction for motion. ICC(3,1) was 0.97 for static scans, 0.97 for dynamic scans uncorrected for motion, and 0.97 for dynamic scans after correction for motion. Reliability coefficients were 1.54 mL for static scans, 2.26 mL for dynamic scans uncorrected for motion, and 3.52 mL for dynamic scans after correction for motion.

## Human participant scans

The laser scanner was also validated using a human participant volunteer. The subject was a 60 year old man (body mass = 109 kg, height = 1.72 meters) with a unilateral transtibial amputation due to diabetes that was performed 2.5 years prior. This research was approved by the Virginia Tech Institutional Review Board (IRB# 20–364), and the subject provided written informed consent prior to participation. Upon arrival, the subject was asked to sit in a chair

**Table 3. Errors in cylinder diameter, volume, and surface area from dynamic scans after motion correction.**

| Calibration | Mean Diameter (mm) | Mean Diameter Error (mm) | Mean Diameter Error (%) | Max Diameter Error (%) | Volumetric Error (%) | Surface Area Error (%) |
|---|---|---|---|---|---|---|
| 1 | 102.35 | 0.101 | 0.099 | 0.398 | 0.33±0.12 | −0.52±0.05 |
| 2 | 102.49 | 0.235 | 0.229 | 0.513 | 0.53±0.16 | −0.37±0.14 |
| 3 | 102.04 | -0.216 | -0.212 | -0.456 | −0.28±0.18 | −0.87±0.17 |

**Table 4. Changes in volumetric and surface area percent errors of dynamic cylinder scans, relative to the initial static scans.**

| Calibration | Static Test Case | | 2 Hz, Uncorrected | | 2 Hz, Motion Corrected | |
|---|---|---|---|---|---|---|
| | Volume Error (%) | Surface Area Error (%) | Relative Volumetric Error (% Change) | Relative Surface Area Error (% Change) | Relative Volumetric Error (% Change) | Relative Surface Area Error (% Change) |
| 1 | 0.18±0.03 | −0.58±0.06 | -0.37 | 0.01 | 0.13 | 0.06 |
| 2 | 0.14±0.19 | −0.65±0.22 | -0.42 | 0.00 | 0.39 | 0.28 |
| 3 | −0.20±0.08 | −0.79±0.08 | -0.15 | -0.11 | -0.08 | -0.08 |

and remove his socket and liner. Following a 20-minute rest period to allow the limb volume to equilibrate, Phase 1 involved five consecutive scans without moving the limb between scans. These scans provided a baseline and measure of limb resting volume. Next, for Phase 2, the subject was asked to remove and reinsert his limb into the scanning area three times, each time taking one scan to give a measure of repeatability of scanning for different initial limb orientations. Next, the subject donned his socket and stood for 15-minutes. Then subject then sat again in the chair, quickly doffed his socket, and immediately inserted his limb into the scanning area. Phase 3 scans were then completed back-to-back for 15 minutes, totaling 12 scans with the first scan beginning approximately 10 seconds after doffing of the socket. This final set of scans was made to demonstrate the laser scanners' ability to measure short-term limb volume changes of a residual limb after removal from a socket. Approximately 130 mm of the residual limb length was scanned (determined by the length of the limb itself and not limited by the laser scanner). Approximately 700,000 data points were collected over the surface of the limb on each scan, and an example of corresponding point cloud and meshed limb surface are shown in Fig 20. Here in Fig 20B, the scanned surface of a dot tracking sticker, with a thickness of 0.11 mm, is visible.

Limb volumes are plotted for all scans during the entire session in Fig 21. Phase 1 scans were used to assess the repeatability of measurements while maintaining the same limb position and orientation across repeated scans. From these scans, the mean limb volume was 1,157 mL, with a standard deviation of 8.13 mL, or 0.7%. Phase 1 volume measurements exhibited a TEM of 5.76 mL and reliability coefficient of 15.96 mL.

Phase 2 scans were used to assess the repeatability of measurements without strictly maintaining the same limb position and orientation across repeated scans (Fig 21). The measured limb volume during these scans was expected to remain within the bound of the Phase 1 scans. This was supported in that the Phase 2 mean limb volume was measured to be 1,151 mL with a

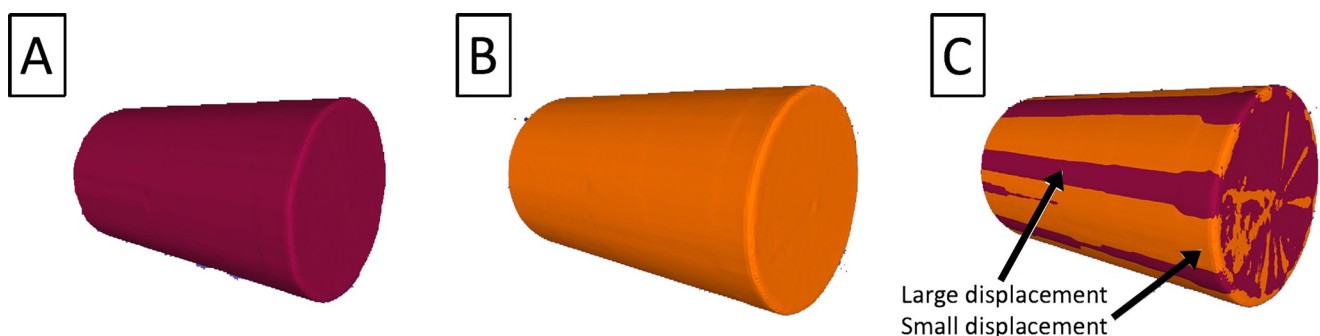

**Fig 19. Displacements of the cylinder during dynamic tests.** (A) Uncorrected meshed scan. (B) Corrected meshed scan. (C) Overlay of A and B, with red bands indicating large positive displacements, and orange bands indicating negative or neutral displacements.

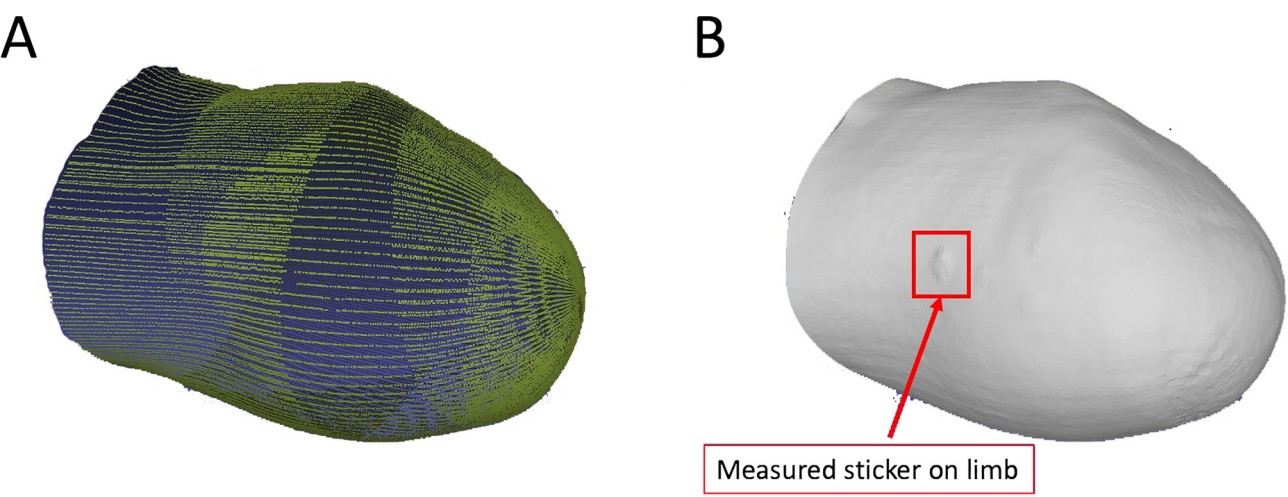

**Fig 20.** (A) Down sampled point cloud data. (B) meshed limb surface for human participant. Note in (B) the scanned surface of a dot tracking sticker.

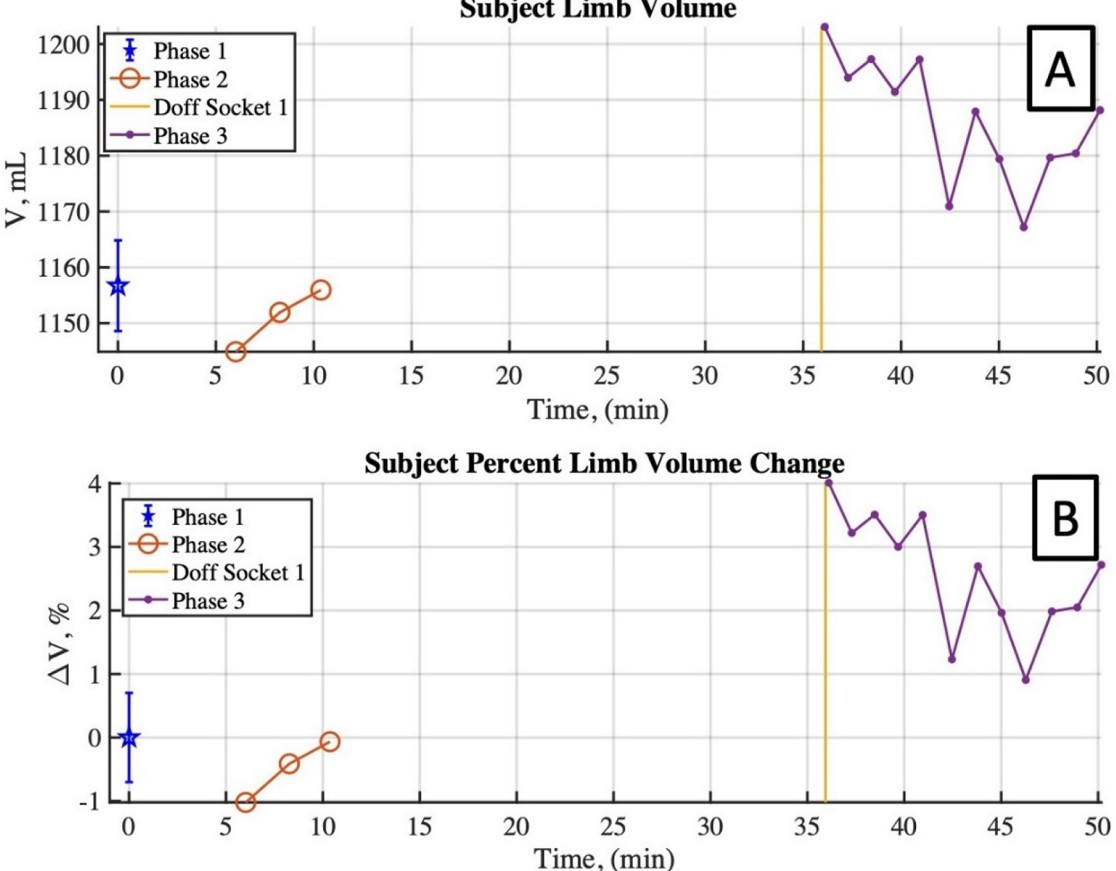

**Fig 21. Human participant limb volume time history over the three scanning phases.** Phase 1 results are combined and reported with error bars. (A) Measured limb volume between scan sets. (B) Measured percent limb volume change relative to Phase 1 results.

standard deviation of 5.6 mL (0.48%), and a mean difference from the Phase 1 scans of −5.79 mL *or* −0.5%. Potential sources of error in these results are 1) out of plane (toward or away from the cameras) limb motion that is not measured and corrected for, and 2) induced errors associated with limb meshing and alignment between meshes in MeshLab. The difficulty in aligning repeated limb surface meshes has been previously acknowledged [34], but there remains little consensus on a robust method for accurately aligning deformable bodies such as residual limbs.

Lastly, Phase 3 limb scans aimed to capture limb volume change immediately following 15 minutes of standing (Fig 21). These volume measurements indicate an initial volume increase of 46 mL (4%) from the Phase 1 mean volume, approximately 23 mL (2%) recovery of this volume over the subsequent 15 minutes. These results demonstrate the limb scanner's ability to measure both limb volume and volume changes associated with physical activity. Similar to Phases 1 and 2, the primary sources of error in the results are believed to be out of plane limb motion and post processing mesh alignment.

The reliability of our volume measurements compared favorably with other scanners reported in the literature. Our ICC of 0.97 during static and dynamic model cylinder scans was similar to the 0.99 reported for the Artec Eva structured light scanner (Artec Group, Luxembourg) [18], a VIUScan marker-assisted laser scanner (Creaform Inc., Levis, Canada) [17], and Go!SCAN structured white light scanner (Creaform Inc.; Levis, Canada) [17]. These values are higher than the 0.90 threshold reported elsewhere for clinically-relevant reliability [26,35]. Our reliability coefficients of 1.54–3.52 mL during static and dynamic model cylinder scans were similar to the 2.16–2.66 mL for the VIUScan and Go!SCAN [17], and smaller than the 101–131 mL reported for Rodin4D O&P laser-line scanner (Pessac, France), TT Design system (Ottobock, Duderstadt, Germany), Biosculptor Bioscanner (Hialeah, FL, USA), and Omega structured light scanner (Mt. Sterling, OH, USA) [33]. While our model was a cylinder, the other studies cited here used realistic-shaped residual limb models, and reliability coefficients can be expected to be larger with models shaped like residual limbs compared to geometric shapes [5]. We considering volume measurements of residual limbs, our reliability coefficient of 15.96 mL during Phase 1 of the human participant scans was smaller than the 67.2 mL for the Artec Eva [26], 70.7 for the Omega scanner [26], 24.3 for the Go!SCAN [22], 26.4 mL for the Rodin4D [23], and 32.8 mL for the Biosculptor [23]. The reliability coefficients for the residual limbs of human participants are typically higher than those of models due to subject movements during scans and varied shapes of the residuum [26]. The scan time for this system is approximately 35 seconds, which is shorter than that reported for the handheld Artec Eva and Omega Scanner 3D [26].

Multiple limitations of the current study are worth noting. First, our human participants sample size of one limited the generalizability of our results across varied residuum sizes and shapes, and precluded calculations of ICC for reliability. Second, data analysis was time-consuming and requires significant technical training and expertise. This, combined with the physical size and necessary calibration of our scanner, likely preclude it from clinical application.

## Conclusions

A novel 3D laser scanning system was developed for the purpose of measuring the volume and shape of transtibial residual limbs. The following are key conclusions from this research.

- A high accuracy laser line scanner was used to collect surface measurements, and a two-camera motion correction system was developed to estimate and correct for rigid body motions

of the limb. Because this system is automated and used motors to move the scanner over the limb to perform a scan, there is no variability in measurements between operators.

- The laser's accuracy and reliability were validated using a 4-inch cylindrical body of known geometry under both static and dynamic test conditions. Multiple laser calibrations were performed, and the inter-calibration and intra-calibration repeatability of the system was assessed. A negligible difference in repeatability was found as a result, lending confidence to the developed calibration method for the system. Static cylinder scan measurements demonstrated a high accuracy cylinder diameter, volume, and surface area. Among the three calibrations, maximum percent errors of these three metrics were found to be 0.144%, -0.2%, and -0.79%, respectively.

- Dynamic testing conditions were completed by exciting the cylinder at 2 Hz using a linear shaker. The same three metrics were used to assess the accuracy and repeatability of scan measurements both with and without motion correction applied to the scan data. The results indicated that the measurement errors can be improved through by applying motion correction, with decreases in volumetric error of up to 0.24% observed.

- Lastly, the residual limb of a transtibial amputee was scanned under three sets of scans: Phase 1: Initial scanning after socket doffing and 20 minutes of rest to allow for limb volume equilibration, Phase 2: Scans following removal and reinsertion of the limb to allow for natural variation in limb orientation, and Phase 3: Rapid scans completed over 15 minutes immediately following a 15 minute period of exercise and subsequent socket doffing.

- The results of these scans demonstrated limb volume standard deviation of 0.7% in Phase 1, with the three Phase 2 scans falling within acceptable bounds compared to the Phase 1 measurements. The 12 Phase 3 scans, following activity and socket doffing, demonstrated a maximum volume increase of 4% following doffing, with a limb volume recovery of 2% compared to Phase 1 after the 15-minute period.

## Author Contributions

**Conceptualization:** Carson O. Squibb, Michael L. Madigan, Michael K. Philen.

**Data curation:** Carson O. Squibb, Michael L. Madigan.

**Formal analysis:** Carson O. Squibb, Michael L. Madigan, Michael K. Philen.

**Funding acquisition:** Michael L. Madigan, Michael K. Philen.

**Investigation:** Michael L. Madigan, Michael K. Philen.

**Methodology:** Carson O. Squibb, Michael K. Philen.

**Project administration:** Michael K. Philen.

**Software:** Carson O. Squibb.

**Supervision:** Michael L. Madigan, Michael K. Philen.

**Validation:** Carson O. Squibb, Michael K. Philen.

**Visualization:** Carson O. Squibb.

**Writing – original draft:** Carson O. Squibb.

**Writing – review & editing:** Carson O. Squibb, Michael L. Madigan, Michael K. Philen.

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
