## [Decision Letter · Decision Letter 0]

19 Oct 2023

PONE-D-23-22780A High Precision Laser Scanning System for Measuring Shape and Volume of Transtibial Amputee Residual Limbs: Design and ValidationPLOS ONE

Dear Dr. Philen,

Thank you for submitting your manuscript to PLOS ONE. After careful consideration, we feel that it has merit but does not fully meet PLOS ONE’s publication criteria as it currently stands. Therefore, we invite you to submit a revised version of the manuscript that addresses the points raised during the review process.

We look forward to receiving your revised manuscript.

Kind regards,

Mohammad Azadi

Academic Editor

PLOS ONE

“This material is based upon work supported by the National Science Foundation under Grant No. 1906132.”

“This material is based upon work supported by the National Science Foundation under Grant No. 1906132.   The funders had no role in study design, data collection and analysis, decision to publish, or preparation of the manuscript.”

Additional Editor Comments:

The manuscript must be revised based on the reviewers’ comments plus the following issues,

1) A separated file must be provided for the authors’ answers to the comments, one by one. Moreover, all changes must be yellow-colored highlighted sentences in the revised article. The track changes condition is not suggested.

2) Using references suggested by the reviewers is not mandatory. If they are related, they could be used by the authors. If not, they could be ignored by the authors.

3) All keywords must be found in the abstract or the title.

4) The novelty of the manuscript must be highlighted in the introduction, compared to the literature review.

5) All formulations need references, unless they were extracted or introduced by the authors.

6) The background color of Figures 15, 19, and 20 must be white. The scale bar must be also added.

7) The digit numbers must be similar in Tables 2 and 4.

8) The conclusion section should be rewritten one by one, in bullets, to show the novelty.

9) The discussion is poor and it must be improved. They must be compared to other results of other similar articles.

10) References should be updated based on recent articles, published in 2015-2023. Moreover, it should be extended to at least 35 articles for a proper discussion.

Reviewers' comments:

Reviewer's Responses to Questions

**Comments to the Author**

1. Is the manuscript technically sound, and do the data support the conclusions?

Reviewer #1: Yes

Reviewer #2: Yes

Reviewer #3: Partly

2. Has the statistical analysis been performed appropriately and rigorously? 

Reviewer #1: Yes

Reviewer #2: Yes

Reviewer #3: No

3. Have the authors made all data underlying the findings in their manuscript fully available?

Reviewer #1: Yes

Reviewer #2: Yes

Reviewer #3: No

4. Is the manuscript presented in an intelligible fashion and written in standard English?

Reviewer #1: Yes

Reviewer #2: Yes

Reviewer #3: Yes

5. Review Comments to the Author

Reviewer #1: The paper focuses on developing and validating a 3D laser scanner for measuring the volume and shape of residual limbs among amputees.

The introduction needs to discuss previous studies on using laser scanning for measuring residual limb shape and volume, along with their limitations.

Make the figure captions more concise while ensuring clarity, for example in Figure 5. Please also check for typos.

Explain the parameters used in Figures within the main text (for example, in Figure 7). Clarify the differences between the flat calibration object and the flat plate.

Explain the reasoning behind choosing a 2Hz excitation frequency for evaluating the motion correction system. Also, clarify why excitation was done in the y-direction.

It is suggested to explore alternative methods to verify result accuracy beyond inter-data variability analysis.

Reviewer #2: This paper presents a nice novel method for automated 3D scanning of a residual limb towards prosthetic socket design, which claims improvements over handheld scanners and shows some novelty over prior rotating/armature-mounted scanners (i.e. not just rotating around limb axis but also covering tip, and spiral path). The authors should be complemented on a very detailed disclosure of method enabling reproducibility.

My main concern is around the many gaps in the literature review meaning that the scientific background is not adequately acknowledged and the clinical need is unclear.

The most recent cited paper was published in 2019 and there has been a lot of work since, in validating hand-held scanners. While hand-held scanners will likely never be as controlled and user-independent as an automated device like the one presented, we do need to consider what is ‘good enough’ and what is practical, low cost, reliable and unintimidating for patients in clinics.

The claimed shortcomings of scanning are based on the paper’s reference [17], around user variability; is this really still the case though? [17] is from 2010 and newest reference is from 2019, which misses much more recent work e.g. DOI:10.1109/TBME.2019.2895283 from 2019 and DOI: 10.1097/PXR.0000000000000105, https://doi.org/10.3390/s22186863 and DOI: 10.1097/JPO.0000000000000350 all from 2022 and tested with living participants, and the latter providing a comparison with hands-on casting for clinically-successful context. These scanners are now showing clinically valid results which are near or better than limit of detectible volume change and better than plaster casting. There are also several missing studies from earlier in the development of CAD/CAM (see e.g. McGarry and colleagues from Strathclyde – not this reviewer ) who did a lot of foundational work on assessment vs. cylindrical phantoms in the 2000s.

Also, a more fundamental question - if showing a new system and method, why not compare its capabilities to a clinically-standard hand-held scanner?

The study is also very light on acknowledged limitations and requirements for future work, especially with a view to clinical application. For example, the present study covers calibration and validation with residual limb analogue objects nicely, and shows a real-life case study, but the above mentioned papers often considered multiple test participants, enabling characterization of the effect of e.g. measurement size upon error or variability. Factors like packaging and development engineering to take this from a research prototype to a clinically usable device, should be noted.

Reviewer #3: Hi,

Evidence for the innovation of the article has not been provided, so it is suggested that a comparison be made with similar researches and the advantages of this method be expressed with them. For example, compare the amount of errors with reference (25. Seminati E, Canepa Talamas D, Young M, Twiste M, Dhokia V, Bilzon JL. Validity and

reliability of a novel 3D scanner for assessment of the shape and volume of amputees’ residual

limb models. PloS one. 2017;12(9):e0184498.).

Statistical analysis is incomplete and the data should be provided to the reader in the form of tables or graphs.

6. PLOS authors have the option to publish the peer review history of their article (what does this mean?). If published, this will include your full peer review and any attached files.

Reviewer #1: No

Reviewer #2: No

Reviewer #3: No

---

## [Author Response · Author response to Decision Letter 0]

8 Jan 2024

>>> The authors would like to thank the reviewers for their time in reviewing the article and their valuable suggestions. We have made every attempt to fully address the concerns noted in the reviews. Below you will find our point-by-point responses to each Editor and reviewer comment. We have indicated our responses by “>>>” and italics. We have also highlighted in yellow any changes to the paper itself as a result of addressing these comments. 

General Editor Comments 

Please remove any funding-related text from the manuscript and let us know how you would like to update your Funding Statement. 

>>> It has been removed from the manuscript.

Let us know how you would like to update your Funding Statement. 

>>> “This material is based upon work supported by the National Science Foundation under Grant No. 1906132. The funders had no role in study design, data collection and analysis, decision to publish, or preparation of the manuscript.”

We note that you have stated that you will provide repository information for your data at acceptance. Should your manuscript be accepted for publication, we will hold it until you provide the relevant accession numbers or DOIs necessary to access your data.

>>> Thank you.

Please include your full ethics statement in the ‘Methods’ section of your manuscript file. In your statement, please include the full name of the IRB or ethics committee who approved or waived your study, as well as whether or not you obtained informed written or verbal consent. If consent was waived for your study, please include this information in your statement as well.

>>> We have added the requested information.

Additional Editor Comments

A separated file must be provided for the authors’ answers to the comments, one by one. Moreover, all changes must be yellow-colored highlighted sentences in the revised article. The track changes condition is not suggested.

>>> We have included our responses to reviewer comments here, and highlighted additions to the article in yellow.

Using references suggested by the reviewers is not mandatory. If they are related, they could be used by the authors. If not, they could be ignored by the authors.

All keywords must be found in the abstract or the title.

>>> We have confirmed that all keywords confirmed can be found in the abstract or title.

The novelty of the manuscript must be highlighted in the introduction, compared to the literature review.

>>> Extra content in the introduction was added that compare the system in this work with previous works.

All formulations need references, unless they were extracted or introduced by the authors.

>>> All formulations were derived by the authors and no references were used.

The background color of Figures 15, 19, and 20 must be white. The scale bar must be also added.

>>> We have changed the background color of these figures to white. We are unsure how to create a color scale bar for the images as the colors (e.g. red, orange) are only used to distinguish between two different scans and have no numerical value. 

The digit numbers must be similar in Tables 2 and 4.

>>> The Table were formatted to be consistent with all other tables. The number of decimal places is consistent throughout the table. 

The conclusion section should be rewritten one by one, in bullets, to show the novelty.

>>> The conclusion has been expanded using a bulleted point list.

The discussion is poor and it must be improved. They must be compared to other results of other similar articles.

>>> We have added measures of reliability of scan volume to our Results, and added to the Discussion a comparison of these results to other studies.

References should be updated based on recent articles, published in 2015-2023. Moreover, it should be extended to at least 35 articles for a proper discussion.

>>> We added the most recent articles on the subject of scanning the residual limb, including a short comment on each article.

Bailey-Brændgaard, M., and Enevoldsen, P. W. "Accuracy and Reliability of 3D Scanning Spatial Data when Capturing Limb Morphology for Use within Prosthetics and Orthotics: A Scoping Review," 2022.

Dickinson, A. S., Donovan-Hall, M. K., Kheng, S., Bou, K., Tech, A., Steer, J. W., Metcalf, C. D., and Worsley, P. R. "Selecting appropriate 3D scanning technologies for prosthetic socket design and transtibial residual limb shape characterization," JPO: Journal of Prosthetics and Orthotics Vol. 34, No. 1, 2022, pp. 33-43.

Kofman, R., Winter, R. E., Emmelot, C. H., Geertzen, J. H., and Dijkstra, P. U. "Clinical usability, reliability, and repeatability of noncontact scanners in measuring residual limb volume in persons with transtibial amputation," Prosthetics and Orthotics International Vol. 46, No. 2, 2022, pp. 164-169.

Ngan, C. C., Sivasambu, H., Ramdial, S., and Andrysek, J. "Evaluating the Reliability of a Shape Capturing Process for Transradial Residual Limb Using a Non-Contact Scanner," Sensors Vol. 22, No. 18, 2022, p. 6863.

Rodrigues, A. S., Oliveira, M. C., and Da Gama, A. E. "Anthropometric Evaluation of Two Low-Cost 3D Digitizers for Lower Limb Prosthetics: A Preliminary Study," 2022.

Seminati, E., Young, M., Canepa Talamas, D., Twiste, M., Dhokia, V., and Bilzon, J. "Reliability of three different methods for assessing amputee residuum shape and volume: 3D scanners vs. circumferential measurements," Prosthetics and Orthotics International Vol. 46, No. 4, 2022, pp. 327-334.

Reviewer #1: 

The paper focuses on developing and validating a 3D laser scanner for measuring the volume and shape of residual limbs among amputees.

The introduction needs to discuss previous studies on using laser scanning for measuring residual limb shape and volume, along with their limitations.

>>> The introduction has been expanded with a discussion on other scanning systems with their limitations.

Make the figure captions more concise while ensuring clarity, for example in Figure 5. Please also check for typos.

>>> Edits were made to the captions to make them more concise.

Explain the parameters used in Figures within the main text (for example, in Figure 7). Clarify the differences between the flat calibration object and the flat plate.

>>> We feel the current explanation of the parameters in the Figure 7 caption is reasonable and prefer to keep them there. We have also added explanations of parameters in the Figure 1 caption.

Explain the reasoning behind choosing a 2Hz excitation frequency for evaluating the motion correction system. Also, clarify why excitation was done in the y-direction.

>>> Sentences were added to the text to explain these two questions.

It is suggested to explore alternative methods to verify result accuracy beyond inter-data variability analysis.

>>> We have added measurements of typical error of the measurement (TEM), intraclass correlation coefficient, and reliability coefficients—all of which have been used by other papers to quantify the performance of various scanner systems. We have also added a comparison of these measurements from our scanner with other scanners reported in the literature. 

Reviewer #2: 

This paper presents a nice novel method for automated 3D scanning of a residual limb towards prosthetic socket design, which claims improvements over handheld scanners and shows some novelty over prior rotating/armature-mounted scanners (i.e. not just rotating around limb axis but also covering tip, and spiral path). The authors should be complemented on a very detailed disclosure of method enabling reproducibility.

My main concern is around the many gaps in the literature review meaning that the scientific background is not adequately acknowledged and the clinical need is unclear.

The most recent cited paper was published in 2019 and there has been a lot of work since, in validating hand-held scanners. While hand-held scanners will likely never be as controlled and user-independent as an automated device like the one presented, we do need to consider what is ‘good enough’ and what is practical, low cost, reliable and unintimidating for patients in clinics.

The claimed shortcomings of scanning are based on the paper’s reference [17], around user variability; is this really still the case though? [17] is from 2010 and newest reference is from 2019, which misses much more recent work e.g. DOI:10.1109/TBME.2019.2895283 from 2019 and DOI: 10.1097/PXR.0000000000000105, https://doi.org/10.3390/s22186863 and DOI: 10.1097/JPO.0000000000000350 all from 2022 and tested with living participants, and the latter providing a comparison with hands-on casting for clinically-successful context. These scanners are now showing clinically valid results which are near or better than limit of detectible volume change and better than plaster casting. There are also several missing studies from earlier in the development of CAD/CAM (see e.g. McGarry and colleagues from Strathclyde – not this reviewer ) who did a lot of foundational work on assessment vs. cylindrical phantoms in the 2000s.

>>> Thank you for these suggested papers. We have updated our Introduction by including more recent work suggested by the reviewer. 

Also, a more fundamental question - if showing a new system and method, why not compare its capabilities to a clinically-standard hand-held scanner?

>>> We agree that this would have been an intriguing addition to the paper. However, the resources to obtain a clinically-standard hand-held scanner were not available. In lieu of this, and based on comments from reviewers 1 and 3, we have added to our Discussion a comparison between the current scanner and others described in the literature.

The study is also very light on acknowledged limitations and requirements for future work, especially with a view to clinical application. For example, the present study covers calibration and validation with residual limb analogue objects nicely, and shows a real-life case study, but the above mentioned papers often considered multiple test participants, enabling characterization of the effect of e.g. measurement size upon error or variability. Factors like packaging and development engineering to take this from a research prototype to a clinically usable device, should be noted.

>>> We have added measures of reliability of scan volume to our Results, and added to the Discussion a comparison of these results to other studies. We have also added to our discussion of limitations and future work.

 

Reviewer #3: 

Hi,

Evidence for the innovation of the article has not been provided, so it is suggested that a comparison be made with similar researches and the advantages of this method be expressed with them. For example, compare the amount of errors with reference (25. Seminati E, Canepa Talamas D, Young M, Twiste M, Dhokia V, Bilzon JL. Validity and

reliability of a novel 3D scanner for assessment of the shape and volume of amputees’ residual

limb models. PloS one. 2017;12(9):e0184498.).

Statistical analysis is incomplete and the data should be provided to the reader in the form of tables or graphs.

>>> We have added measures of reliability of scan volume to our Results, and added to the Discussion a comparison of these results to other studies. Thank you for the suggested reference. We have added this to our manuscript and compared our results to it.

---

## [Decision Letter · Decision Letter 1]

29 Jan 2024

PONE-D-23-22780R1A High Precision Laser Scanning System for Measuring Shape and Volume of Transtibial Amputee Residual Limbs: Design and ValidationPLOS ONE

Dear Dr. Philen,

Thank you for submitting your manuscript to PLOS ONE. After careful consideration, we feel that it has merit but does not fully meet PLOS ONE’s publication criteria as it currently stands. Therefore, we invite you to submit a revised version of the manuscript that addresses the points raised during the review process.

We look forward to receiving your revised manuscript.

Kind regards,

Mohammad Azadi

Academic Editor

PLOS ONE

Journal Requirements:

Additional Editor Comments:

Some minor revisions must be done, as follows,

1) The additions at L112-3 are better suited to the Discussion.

2) Instead of 'human subject' suggest 'human participant' (more inclusive).

3) Again, please check all formulations. They need references, unless they were extracted by the authors.

4) The structure is confusing. The text must have an introduction, research method, results and discussion, conclusions, and references. Others must be subparts of these sections.

5) In conclusions, a general sentence must be added about the topic.

6) Two keywords of "residual limb volume" and "laser scanner" could not be found in the abstract or the title.

7) The discussion was added in the last part. For the first parts of results, no discussion was added. It is better to add the discussion with new references to extend them up to at least 35 articles, published in 2023-2024. No references could be found for 2023.

Reviewers' comments:

Reviewer's Responses to Questions

**Comments to the Author**

1. If the authors have adequately addressed your comments raised in a previous round of review and you feel that this manuscript is now acceptable for publication, you may indicate that here to bypass the “Comments to the Author” section, enter your conflict of interest statement in the “Confidential to Editor” section, and submit your "Accept" recommendation.

Reviewer #1: All comments have been addressed

Reviewer #2: All comments have been addressed

Reviewer #3: All comments have been addressed

2. Is the manuscript technically sound, and do the data support the conclusions?

Reviewer #1: Yes

Reviewer #2: Yes

Reviewer #3: Yes

3. Has the statistical analysis been performed appropriately and rigorously? 

Reviewer #1: Yes

Reviewer #2: Yes

Reviewer #3: Yes

4. Have the authors made all data underlying the findings in their manuscript fully available?

Reviewer #1: (No Response)

Reviewer #2: Yes

Reviewer #3: Yes

5. Is the manuscript presented in an intelligible fashion and written in standard English?

Reviewer #1: Yes

Reviewer #2: Yes

Reviewer #3: Yes

6. Review Comments to the Author

Reviewer #1: The manuscript titled 'A High Precision Laser Scanning System for Measuring Shape and Volume

of Transtibial Amputee Residual Limbs: Design and Validation' can be accepted for the publication.

Reviewer #2: Thanks for careful consideration of my previous comments. I think the paper looks useful to the community and should be shared.

Very minor points that do not require another view from me:

* additions at L112-3 are better suited to the Discussion.

* instead of 'human subject' suggest 'human participant' (more inclusive)

Reviewer #3: (No Response)

7. PLOS authors have the option to publish the peer review history of their article (what does this mean?). If published, this will include your full peer review and any attached files.

Reviewer #1: No

Reviewer #2: No

Reviewer #3: **Yes: **Seyed Morteza Hosseini

---

## [Author Response · Author response to Decision Letter 1]

5 Mar 2024

>>> The authors would like to thank the reviewers for their time in reviewing the article and their valuable suggestions. We have made every attempt to fully address the concerns noted in the reviews. Below you will find our point-by-point responses to each Editor and reviewer comment. We have indicated our responses by “>>>” and italics. We have also highlighted in yellow any changes to the paper itself as a result of addressing these comments. 

General Editor Comments 

1) The additions at L112-3 are better suited to the Discussion. 

>>> These have been moved to the Discussion.

2) Instead of 'human subject' suggest 'human participant' (more inclusive).

>>> These changes have been made.

3) Again, please check all formulations. They need references, unless they were extracted by the authors.

>>> All formulations were checked. These were derived by the first author Carson Squibb and no references were used in the derivation. 

4) The structure is confusing. The text must have an introduction, research method, results and discussion, conclusions, and references. Others must be subparts of these sections.

>>> The structure did follow the suggestions; however the “Method” section was changed to “Research Method” as suggested. The subparts sections have been confirmed to be Heading 2 where as the main sections are Heading 1 style.

5) In conclusions, a general sentence must be added about the topic.

>>> A general sentence was added to the conclusion before the bulleted key conclusions. 

6) Two keywords of "residual limb volume" and "laser scanner" could not be found in the abstract or the title.

>>> Residual limb volume was added to the abstract. Laser scanner was confirmed to be in the abstract.

7) The discussion was added in the last part. For the first parts of results, no discussion was added. It is better to add the discussion with new references to extend them up to at least 35 articles, published in 2023-2024. No references could be found for 2023.

>>> The first parts of our Results report calibration parameters that are specific to our particular scanner and thus we felt require no Discussion or comparison with other studies. The remaining parts of our Results report on static cylinder tests, dynamic cylinder tests, scanner reliability, and human participant scans. All of these were discussed and compared to other studies in the latter part of our Results and Discussion section (Lines 591-610).

>>> The following two references published in 2023 were added to the introduction. While these do not focus on measuring volume/shape changes in the residual limb, they do focus on using 3D scanners for measuring leg volume and socket shape. 

27. Ashby, J., Lewis, M., Sunderland, C., Barrett, L. A., and Morris, J. G. "The Reliability and Validity of a Portable Three-Dimensional Scanning System to Measure Leg Volume," Sensors Vol. 23, No. 22, 2023, p. 9177.

28. Nagarajan, Y. R., Farukh, F., Silberschmidt, V. V., Kandan, K., Singh, A. K., and Mukul, P. "Shape Analysis of Prosthetic Socket Rectification Procedure for Transtibial Amputees," Prosthesis Vol. 6, No. 1, 2024, pp. 157-174

---

## [Editor Report · Decision Letter 2]

19 Mar 2024

A High Precision Laser Scanning System for Measuring Shape and Volume of Transtibial Amputee Residual Limbs: Design and Validation

PONE-D-23-22780R2

Dear Dr. Philen,

We’re pleased to inform you that your manuscript has been judged scientifically suitable for publication and will be formally accepted for publication once it meets all outstanding technical requirements.

An invoice for payment will follow shortly after the formal acceptance. To ensure an efficient process, please log into Editorial Manager at Editorial Manager® , click the 'Update My Information' link at the top of the page, and double check that your user information is up-to-date. If you have any billing related questions, please contact our Author Billing department directly at authorbilling@plos.org.

Kind regards,

Mohammad Azadi

Academic Editor

PLOS ONE

Additional Editor Comments (optional):

Almost done!
---

## [Editor Report · Acceptance letter]

5 Apr 2024

PONE-D-23-22780R2 

PLOS ONE

Dear Dr. Philen, 

I'm pleased to inform you that your manuscript has been deemed suitable for publication in PLOS ONE. Congratulations! Your manuscript is now being handed over to our production team.

Kind regards, 

on behalf of

Dr. Mohammad Azadi 

Academic Editor

PLOS ONE